# Epidemiology of Mucopolysaccharidoses Update

**DOI:** 10.3390/diagnostics11020273

**Published:** 2021-02-10

**Authors:** Betul Celik, Saori C. Tomatsu, Shunji Tomatsu, Shaukat A. Khan

**Affiliations:** 1Nemours/Alfred I. duPont Hospital for Children, Wilmington, DE 19803, USA; duzleme@udel.edu (B.C.); stomatsu@nemours.org (S.T.); 2Department of Biological Sciences, University of Delaware, Newark, DE 19716, USA; stomatsu@udel.edu

**Keywords:** mucopolysaccharidoses, glycosaminoglycans, incidence and prevalence, epidemiology, mutations, newborn screening

## Abstract

Mucopolysaccharidoses (MPS) are a group of lysosomal storage disorders caused by a lysosomal enzyme deficiency or malfunction, which leads to the accumulation of glycosaminoglycans in tissues and organs. If not treated at an early stage, patients have various health problems, affecting their quality of life and life-span. Two therapeutic options for MPS are widely used in practice: enzyme replacement therapy and hematopoietic stem cell transplantation. However, early diagnosis of MPS is crucial, as treatment may be too late to reverse or ameliorate the disease progress. It has been noted that the prevalence of MPS and each subtype varies based on geographic regions and/or ethnic background. Each type of MPS is caused by a wide range of the mutational spectrum, mainly missense mutations. Some mutations were derived from the common founder effect. In the previous study, Khan et al. 2018 have reported the epidemiology of MPS from 22 countries and 16 regions. In this study, we aimed to update the prevalence of MPS across the world. We have collected and investigated 189 publications related to the prevalence of MPS via PubMed as of December 2020. In total, data from 33 countries and 23 regions were compiled and analyzed. Saudi Arabia provided the highest frequency of overall MPS because of regional or consanguineous marriages (or founder effect), followed by Portugal, Brazil, the Netherlands, and Australia. The newborn screening is an efficient and early diagnosis for MPS. MPS I has been approved for newborn screening in the United States. After the newborn screening of MPS I, the frequency of MPS I increased, compared with the past incidence rates. Overall, we conclude that the current identification methods are not enough to recognize all MPS patients, leading to an inaccurate incidence and status. Differences in ethnic background and/or founder effects impact on the frequency of MPS, which affects the prevalence of MPS. Two-tier newborn screening has accelerated early recognition of MPS I, providing an accurate incidence of patients.

## 1. Introduction

The mucopolysaccharidoses (MPS) are a group of inherited metabolic disorders caused by the deficiency of lysosomal enzymes that degrade glycosaminoglycans (GAGs). There are seven types of MPS, depending on the deficiency of the lysosomal enzyme. MPS are inherited by autosomal recessive genes, except for MPS II, which is transmitted by the X-linked gene. Depending on the enzyme deficiency, specific GAG(s) are accumulated in different cells, tissues, and organs, which result in complicated clinical ramifications ranging from central nervous system involvement to multi organ failure [1,2,3]. GAGs are the primary storage materials that serve as diagnostic biomarkers for MPS. Therefore, the determination of GAG level, as well as the enzyme activity (the deficient enzymes shown in Table 1) and genotyping, is crucial for the diagnosis of MPS [2]. Each GAG plays its physiological role and is expressed in specific tissues [4]. The evaluation of GAG levels will provide a diagnostic and prognostic biomarker to predict the course of the disease. Each type of MPS has a broad range of clinical manifestations [5,6]. Early recognition of MPS remains an unmet challenge since the clinical phenotype looks normal at birth. Thus, recognizing MPS remains unnoticed in most patients until the disease has progressed and is prominent [6,7,8]. Early diagnosis of the disease provides more therapeutic options and proper management for the patients [7,9] that provides a better quality of life and slows or prevents irreversible damage [10]. 

It is important to understand that the incidence of MPS is affected by prenatal diagnosis. There is a considerable recurrence risk in families with a child with MPS. Therefore, prenatal testing and/or preimplantation genetic testing prevents the birth of following children with MPS in MPS families. In a study conducted by Qubbaj et al. (2008), the family, who had three affected children with a homozygous mutation, p.W159C (p.Trp159Cys), in the GALNS gene, received in vitro fertilization treatment (IVF) and preimplantation genetic diagnosis (PGD). The whole genome amplification was carried out for single blastomeres by multiple displacement amplification (MDA). One embryo with heterozygosity and two with homozygosity were determined, while the other three embryos were genetically normal. These normal embryos were confirmed with the short tandem repeats (STR) analysis, two of which were transplanted to the mother. After childbearing, the infant was tested for the mutation in the GALNS gene. The results showed that the baby was a carrier of this mutation, with normal and healthy growth and development without any sign or symptom for MPS IVA [11]. Altarescu et al. (2011) studied PGD for two families with a severe form of MPS II and one family with both MPS II and albinism. All three families underwent IVF treatment, and polar body biopsy (PB) and intracytoplasmic sperm injection were undertaken. After PB-PGD cycles, the first family had a healthy girl, the second family had healthy twins, and the third family had healthy twins [12]. These prenatal events may reduce the incidence of MPS over the years. 

It is essential to be aware of early diagnostic steps for therapeutic efficacy, along with the genetic history of the disease [13]. Most MPS patients experience severe and progressive signs and symptoms with time [10,13,14]. If untreated, these patients mostly die within a few decades [15]. With the help of developments in technology, effective treatment methods have been implicated in clinical practice. Currently, enzyme replacement therapy (ERT) and hematopoietic stem cell transplantation (HSCT) have been used for patients with several types of MPS (Table 1). Studies have indicated that ERT and HSCT are well tolerated in MPS patients without serious adverse effects and could slow the disease progress [16,17]. Supportive treatments are also utilized, including pain management, insomnia treatment, anti-inflammatory drugs, oxygen therapy, psychomotor therapy, and surgical interventions. However, the therapeutic efficiency of these options has been influenced by various factors, including MPS types, the severity of the disease, novel mutations, age, socioeconomic status, etc. The availability of ERT is limited in practice because of the high cost, limited penetration of blood–brain barrier (BBB), cardiac valves and avascular bone region, and weekly or bi-weekly injection. HSCT is based on the transplantation of healthy donor cells and is reasonably priced compared to ERT; however, the limitations include appropriate donors, mortality risk in some procedures, and well-trained facility [15,18]. Other therapeutic options are under development, such as gene therapy, substrate reduction therapy, and pharmacological chaperon therapy. It is critical to know the population study on MPS patients to predict the incidence of affected newborns and the number of patients to be treated. In the previous study, Khan et al. (2018) reported the epidemiology of MPS from 22 countries and 16 regions. The estimated prevalence of MPS has been influenced by the fact that there is a significant variation among nations and ethnic backgrounds, depending on which type of MPS dominates a geographic area. The estimated prevalence of MPS patients in various countries has been reported [1,6,19,20,21,22,23,24]. However, these data did not cover the epidemiology of MPS worldwide. The increased number of identified patients, the variation of clinical severity, the identification of novel mutations, insufficient treatment, and screening options have revealed new outcomes. In this study, we aimed to investigate the incidence and prevalence of all MPS types in 33 countries and 23 regions, including the data of Khan et al. (2018), to compare the current MPS situation in the reported countries and 58 new articles researching the frequency of MPS, newborn screening, and mutations.

## 2. Updated Epidemiology of MPS in the Seven Countries

### 2.1. Prevalence of MPS in Brazil

In Brazil, 1652 patients were identified for MPS from 1982 to 2019. The data were collected from the Medical Genetics Service of Hospital de Clinicas Porto Alegre. A total of 74,215,086 live births were recorded in the Information System on Live Births of the Brazilian Health System Database. The incidence of MPS during this period was 1.57 per 100,000 live births. The incidence per 100,000 live births for each MPS type was as follows: MPS I (0.29), MPS II (0.48), MPS IIIA (0.08), MPS IIIB (0.12), MPS IIIC (0.07), MPS IIID (0.001), MPS IVA (0.15), MPS IVB (0.003), MPS VI (0.35), MPS VII (0.02), and MPS IX (0). The most common type among all MPS was MPS II, accounting for 29.84%, followed by MPS VI (21.25%), and other types were determined as MPS I (19.07%), MPS III (16.16%), MPS IV (12.41%), and MPS VII (1.27%). Subtypes of MPS III were identified for 95.50% of the patients; MPS IIIB had the highest frequency with 45.49%, while MPS IIIA and MPS IIIB had frequencies of 26.67% and 27.45%, respectively. Only one patient was identified for MPS IIID (0.39%). Subtypes of MPS IV were analyzed in 96.09% of the patients; MPS IVA was the most frequent type at 96.45%, and the frequency of MPS IVB was 3.55%. In this study, the birth prevalence of MPS per Brazilian region was also calculated. The most significant proportion of patients were from the Southeast (476), followed by the Northeast (379). The South, Center-West, and North part of the country accounted for 177, 73, and 65 patients, respectively. The MPS incidence rates for each region were determined, which were Northeast (1.78), South (1.66), Southeast (1.62), Center-West (1.26), and North (0.88). Within the MPS groups, MPS II was the highest in all Brazilian regions, which were Northeast (0.51), South and Southeast (0.49), Center-West (0.47), and North (0.38) [25] (Table 2 and Table 3).

### 2.2. Prevalence of MPS in Colombia (Cundinamarca and Boyacá)

In Colombia, the diagnosis of MPS was made between 1998 and 2007. In this study, the data were collected from 11 municipalities of Cundinamarca, and the research methods were concentrated on the evaluation of spatial aggregations of diseases. The results revealed 35 patients with MPS; the most common type was MPS IV, with a frequency of 0.68 per 100,000 newborns. Furthermore, MPS I and MPS II had the same frequency rate of 0.45 per 100,000 newborns, followed by MPS VI with 0.23 per 100,000 newborns. MPS III was the least common type in this period, with a frequency of 0.17 per 100,000 newborns. Finally, the combined frequency of MPS in Colombian province was 1.98 per 100,000 newborns [24]. In another study between 1995 and 2016, 3834 out of 32,940 subjects were suspected of lysosomal storage diseases (LSDs). The urine and blood were investigated for these subjects between 1995 and 2016 [26]. Only 30 subjects were identified for LSDs, of whom 19 patients were diagnosed as MPS. A total of 29,106 subjects were reported between 2005 and 2016, and dried blood spots (DBSs) were examined. Only 622 LDS subjects were found. Among all LSDs, there were 319 MPS patients. The most frequent was MPS IVA, which accounted for 192 patients. MPS VII was diagnosed in only one patient. MPS I, II, III, IVB, and VI patients identified were 27, 43, 14, 2, and 40, respectively [26] (Table 2).

### 2.3. Prevalence of MPS in Malaysia

The prevalence of MPS in Malaysia was calculated between 2003 and 2018, and a total of 83 patients were diagnosed with MPS. All data were obtained from the National Referral Center, Genetic Department, Hospital Kuala Lumpur. Patients were diagnosed by dimethyl methylene blue (DMB) method, high-resolution electrophoresis (HRE) of urinary GAGs (uGAGs), and enzyme assay in plasma or leukocytes. MPS II was the most frequent type to be diagnosed (at 39.6%) among all MPS types. The other MPS types recorded in this country were MPS I (13.3%), MPS III (13.3%), MPS IVA (16.9%), MPS VI (14.5%), and MPS VII (2.4%), with almost all having the severe phenotype [27]. Another study in 2017 by Ngu and colleagues indicated high MPS II type frequency. Out of 79 MPS patients, a total of 30 patients were identified as MPS II [28].

Omar et al. revealed that the overall prevalence of MPS among Malaysian patients at high-risk was 26%. MPS VI accounted for 40% of all MPS and was the most frequent, followed by MPS II at 26.7%, and no case of MPS VII was detected between 2014 and 2016. The samples were collected from all hospitals in Malaysia based on the inclusion criteria. All samples were processed according to the DMB method for uGAGs, and enzyme assay in plasma or leukocytes. A total of 15 cases out of 60 suspected patients were diagnosed with MPS I (one case), MPS II (four cases), MPS IIIA (three cases), MPS IVA (one case), and MPS VI (six cases) [29] (Table 3).

### 2.4. Prevalence of MPS in Mexico

In Mexico, 198 MPS patients from 505 suspected individuals or relatives were diagnosed between 2012 and 2017, with a combined estimated incidence of 2.23 per 100,000 live births. Samples were collected virtually from 32 states of Mexico, and the diagnosis of all patients was made enzymatically. MPS IVA had the highest frequency (49.10%) among all MPS types, while MPS II was the least common type with a frequency of 6.7%. The estimated incidence per 100,000 live births for each MPS type was as follows: MPS I (0.19), MPS II (0.15), MPS IIIA (0.26), MPS IIIB (0.13), MPS IVA (1.10), MPS VI (0.17), and MPS VII (0.23) [30] (Table 2 and Table 3).

### 2.5. Prevalence of MPS in Pakistan

The diagnosis of MPS in Pakistan was made between January 2013 and December 2015. The data were collected from the Children’s Hospital, and Institute of Child Health, Department of Pediatric Gastroenterology, Hepatology and Nutrition. The diagnosis of all patients was confirmed by uGAG assay and enzyme activity assay. A total of 90 out of 195 suspected patients were identified with MPS, which comprised 46.15% of all LSDs. The frequency of MPS I accounted for 83.33% of all MPS, which indicated that MPS I had the highest frequency among all MPS types. Additionally, the other types of identified MPS were MPS II (1.11%), MPS III (5.56%), MPS IV (6.67%), and MPS VI (3.33%). No case of MPS VII was recorded [17] (Table 3).

### 2.6. Prevalence of MPS in Taiwan

In a retrospective study in Taiwan between 1996 and 2017, 28 patients were identified with MPS III. Within the MPS III group, MPS IIIB had the highest number, which accounted for 86%. The frequency of MPS IIIA and MPS IIIC were 11% and 3%, respectively. No case of MPS IIID was identified during the study period [31]. In a previous study conducted by Lin et al., from 1984 to 2004, a total of 6,337,299 live births were recorded. Blood samples were collected from suspected patients to be confirmed by two-dimensional electrophoresis of uGAGs and/or enzyme assay in serum, leukocytes, and/or fibroblasts. One hundred thirty patients were diagnosed with MPS, and the combined birth prevalence was calculated to be 2.04 per 100,000 live births. The highest birth prevalence was recorded for MPS II, with 1.07 per 100,000 live births. However, other MPS types had a low frequency that included MPS I (0.11), MPS III (0.39), MPS IV (0.33), and MPS VI (0.14) [23]. Moreover, in another epidemiological study for MPS disorders conducted in the Taiwanese population, 175 patients with MPS diagnosed from 1985 to 2019 were evaluated regarding the survival and diagnostic age. MPS II, which had the highest frequency, accounted for 45% (78 patients) of all patients, followed by MPS I (12%), MPS III (19%), MPS IV (18%), and MPS VI (6%). Within the MPS III group, the most frequent type was MPS IIIB (15%), followed by MPS IIIA (3%) and MPS IIIC (1%). No patient was recorded for MPS IIID. As of MPS IV, the highest frequency was observed for MPS IVA, which was 18%, and only one patient accounted for MPS IVB (1%). In addition to that, this study indicated that the life expectancy of patients in Taiwan increased compared to the lifespan of patients and also, depending on the newborn screening developments, the early diagnosis has been successfully implemented in this country [32] (Table 2 and Table 3).

### 2.7. Prevalence of MPS in Turkey

In a retrospective study in Turkey, 27 MPS patients were identified at Dokuz Eylul University, Department of Pediatric Metabolism and Nutrition. The diagnosis was confirmed by uGAG assay, specific enzyme level, molecular analysis, and clinical and imaging findings. MPS III, which had the highest frequency, comprised 48% of all MPS, followed by MPS II (26%). Other types of MPS included in this study were MPS I, MPS IV, and MPS VI, with frequencies of 7%, 4%, and 15%, respectively [33].

According to another study conducted in Turkey, out of 836 suspected patients, 45 patients were identified as MPS patients. Urine samples were collected by Hacettepe University Hospitals, and total uGAG was measured by using DMB. MPS III, IV, and VI each accounted for 13 patients, while MPS I and II accounted for two and four patients, respectively [34].

Church et al. measured uGAGs from 530 suspected subjects and confirmed 339 patients with MPS from 2009 to 2011. The most common type was MPS IVA, which involved 21% of all MPS patients. Other types observed in this study were MPS I (6%), MPS II (11%), MPS IIIA (10%), MPS IIIB (9%), MPS IIIC (3%), MPS IVB (1%), and MPS VI (15%), as well as I-Cell (10%) and other LSDs (13%) [35] (Table 3).

### 2.8. Prevalence of MPS in Other Countries

#### 2.8.1. Japan

The birth prevalence of MPS in Japan was investigated from 1982 to 2009. Three hundred thirty-one patients were diagnosed by a semi-quantitative screening of uGAGs with the help of the acid carbazole method between 1982 and 1999, and 136 cases were identified by urine analysis from 2003 to 2009. The combined birth prevalence of MPS cases between 1982 and 2009 accounted for 1.53 per 100,000 live births. Within the MPS group, the birth prevalence of MPS I, II, III, IV, VI, and VII was 0.23, 0.84, 0.26, 0.15, 0.03, and 0.02 per 100,000 live births, respectively. MPS II comprised more than half of all MPS, while MPS I, MPS III, MPS IV, MPS VI, and MPS VII accounted for 15%, 16%, 10%, 1.7%, and 1.3%, respectively [1] (Table 2).

#### 2.8.2. Switzerland

The birth prevalence of MPS in Switzerland was retrospectively determined between 1975 and 2008. uGAG analysis was conducted; however, the results were not sufficient for the final diagnosis. Patients were diagnosed either by measuring the enzyme activity in leukocytes or fibroblasts and/or by molecular analysis. Over 34 years, a total of 2,621,036 live births were recorded, 51 of whom were diagnosed with MPS. The combined birth prevalence was 1.56 per 100,000 live births, and MPS II had the highest birth prevalence of 0.46 per 100,000 live births. Birth prevalence of other MPS types were calculated as 0.19 for MPS I, 0.38 for MPS III, 0.38 for MPS IV, 0.11 for MPS IV, and 0.038 for MPS VII. MPS II accounted for 29% of all MPS, whereas MPS III, MPS IV, and MPS I comprised 24%, 24%, and 12%, respectively. MPS VII had the lowest percentage in Switzerland [1] (Table 2).

#### 2.8.3. Saudi Arabia

The diagnosis of MPS in Saudi Arabia was made between 1983 and 2008. The total number of live births was 165,130 during the study period, and the number of patients diagnosed with MPS was 28. MPS VI had the highest birth prevalence of 7.85 per 100,000 live births, comprising 46% of all MPS cases diagnosed. MPS I and IV accounted for 21% each among all MPS. The birth prevalence of MPS I and MPS VI was 3.62 per 100,000 live births. MPS III was calculated to be 1.8 per 100,000 live births, which accounted for 11% of all cases. No cases of MPS II and VII were detected in Saudi Arabia during this period [36] (Table 2 and Table 3).

#### 2.8.4. South Korea

The birth prevalence of MPS in South Korea was evaluated from 1994 to 2013. The combined birth prevalence of all MPS was calculated to be 1.35 per 100,000 live births. Within the MPS, MPS II had the highest birth prevalence of 0.74 per 100,000 live births, while the birth prevalence of MPS I, III, IV, and VI was 0.21, 0.25, 0.13, and 0.019, respectively. The most common type was MPS II (54.6%), followed by MPS III (18.4%), MPS I (15.3%), and MPS IV (9.5%). No case of MPS VII was reported [37] (Table 2 and Table 3).

#### 2.8.5. China

In an LSD study conducted in China from 2006 to 2012, 376 cases were diagnosed for 17 different LSDs, 50.5% of which belonged to the MPS group. Samples were collected from 21 provinces and municipalities in China, and the most significant proportion of samples was collected from Eastern China (85.8%, n = 322). A small proportion of patients were from Northern China (2), Northeast China (3), Southwest China (8), Northwest China (9), Southern China (11), and Central China (21). MPS II had the highest percentage (47.4%) among all MPS, followed by MPS IVA (26.8%), MPS I (16.3%), MPS VI (4.2%), MPS IIIA/B (3.7%), and MPS VII (1.1%). During the study period, only one patient was diagnosed with MPS IVB [38] (Table 3).

#### 2.8.6. India

In an LSD study conducted in Indian states and two neighboring countries, including Sri Lanka and Afghanistan, between 2002 and 2012. 1110 children were diagnosed with LSD, the majority of whom were glycolipid disorders (48%). Children were from Western part (84.50%; n = 938), Southern (10.9%; n = 121), Northern (2.7%; n = 30), Eastern (0.09%; n = 1), Central parts (0.09%; n = 1), Sri Lanka (1.62%; n = 18), and Afghanistan (0.09%; n = 1). In this study, MPS accounted for 22% (n = 85) of all LSDs. The most common type of MPS was MPS IV (26%), followed by MPS I (25%), MPS VI (21%), MPS III (12%), MPS II (11%), and MPS VII (6%) [39] (Table 3).

#### 2.8.7. Tunisia

The diagnosis of MPS in Tunisia was made from 1988 to 2005. Ninety-six cases of 132 suspected MPS were confirmed by uGAG analysis and/or enzymatic assay for MPS. Only 76 of them were included in the calculation of birth prevalence due to the fact that 14 families were determined by molecular analysis. The total number of live births during that period was 3,309,091. The combined birth prevalence of all MPS was 2.27 per 100,000 live births. MPS III (0.7) had the highest birth prevalence among all MPS, which comprised 32% of all cases. The birth prevalence of MPS I, II, IV, and VI was 0.63, 0.29 (live male births), 0.45, and 0.30, respectively, the percentage of which was 25%, 8%, 21%, and 13% of all MPS. No MPS VII case was reported [40] (Table 2).

#### 2.8.8. Australia

A total of 188 MPS cases were diagnosed between 1980 and 1996. The birth prevalence of MPS III was the highest (1.51 per 100,000 live births, which comprised 35% of all MPS), followed by MPS I (1.14 per 100,000 live births), which accounted for 26% of all MPS. The birth prevalence of MPS II, MPS IVA, MPS VI, and MPS VII were 0.74, 0.59, 0.43, and 0.047 per 100,000 live births. The combined birth prevalence of all MPS was 4.46 per 100,000 live births [20]. In another study conducted in Western Australia between 1969 and 1996, a total of 22 MPS cases were diagnosed, the combined birth prevalence of which was 3.43 per 100,000 live births. The highest birth prevalence was associated with MPS III, which was 1.71 and accounted for 50% of all MPS. The birth prevalence of MPS I, II, IVA, and VI was 0.94 (27%), 0.31 (0.61 for live male births; 9%), 0.31 (9%), and 0.16 (4.5%). There was no case reported for MPS VII [41] (Table 2 and Table 3).

#### 2.8.9. British Columbia (Canada)

The diagnosis of MPS in British Columbia, Canada, was first conducted in 1971 by Lowry and Renwick and then updated by Applegarth et al. with the results obtained from 1969 to 1996. Twenty MPS patients were diagnosed with MPS, with a combined birth prevalence of 1.94 per 100,000 live births. The highest birth prevalence was 0.58 per 100,000 live births for MPS I and comprised 30% of all MPS cases. The birth prevalence of each MPS III, VI, and VII was 0.29, accounting for 15% of all cases, while MPS II and IV were 0.10 (0.19 per 100,000 live male births) and 0.39 per 100,000 live births, comprising 5% and 20% of all cases [42,43,44] (Table 2 and Table 3).

#### 2.8.10. Czech Republic

The Institute of Inherited Metabolic Disorders in Prague reported 4,261,897 live births between 1975 and 2008. A total of 478 cases of LSD were diagnosed in this period, 119 of whom were identified with MPS. The combined birth prevalence of all MPS was 3.72 per 100,000 live births. MPS III had the highest birth prevalence within the MPS group, which was 0.91 per 100,000 live births, accounting for 20% of all MPS cases. The birth prevalence of other MPS types was calculated as MPS I (0.72), MPS II (0.43 for total; 0.83 for male), MPS IV (0.73), MPS VI (0.05), and MPS VII (0.02). The percentage of MPS I, MPS II, MPS IV, MPS VI, and MPS VII were 17%, 18%, 13%, 2%, and 1%, respectively [45] (Table 2 and Table 3).

#### 2.8.11. Denmark

MPS cases were diagnosed at the Kennedy Institute in Glostrup and the Department of Clinical Genetics at Rigshospitalet, Copenhagen, between 1975 and 2004 (30 years). A total of 33 cases were diagnosed with a combined birth prevalence of 1.77 per 100,000 live births. The birth prevalence of other MPS was calculated: MPS I (0.54), MPS II (0.27), MPS III (0.43), MPS IV (0.48), and MPS VI (0.05). The percentage of MPS I, II, III, IV, and VI accounted for 30%, 15%, 24%, 27%, and 3% of all cases. There was no case reported for MPS VII [46] (Table 2 and Table 3).

#### 2.8.12. Norway

The diagnosis of MPS in Norway was conducted at the Department of Clinical Chemistry, Neurochemistry in Molndal, and Genetics at Rikshospitalet, Oslo, from 1979 to 2004. A total of 45 MPS patients were recorded, and the combined birth prevalence was calculated as 3.08 per 100,000 live births. MPS I comprised 60% of all MPS cases, the birth prevalence of which was 1.85 per 100,000 live births. The birth prevalence of MPS II, III, IV, and VI was 0.13, 0.27, 0.76, and 0.07 per 100,000 live births that accounted for 4%, 9%, 24%, and 2% of all cases, respectively [46] (Table 2 and Table 3).

#### 2.8.13. Sweden

MPS suspected samples were collected and diagnosed for 30 years between 1975 and 2004 by the Center for Inborn Errors of Metabolism, Karolinska University Hospital, Huddinge, Stockholm, and Department of Clinical Chemistry and Neurochemistry in Molndal, Sahlgrenska University Hospital, Gothenburg. A total of 52 cases were diagnosed with MPS with a combined birth prevalence of 1.75 per 100,000 live births. The birth prevalence of each MPS I (38%) and MPS III (38%) was 0.67 per 100,000 live births, the highest prevalence rate within the MPS group. The birth prevalence of MPS II, IV, and VI was 0.27, 0.07, and 0.07, accounting for 15%, 4%, and 4% of all cases. No case of MPS VII was reported [46] (Table 2 and Table 3).

#### 2.8.14. Estonia

The diagnosis of MPS was performed by Children’s Hospital in Tallinn and the Department of Genetics of Tartu University Hospital between 1985 and 2006. Toluidine blue spot test was used to select MPS, followed by a quantitative analysis of GAGs in the urine. During this period, 15 MPS cases were identified, and a total of 370,298 live births were recorded. The combined birth prevalence was 4.05 per 100,000 live births. The highest birth prevalence accounted for 2.16 (4.2 per 100,000 live male births) for MPS II, followed by MPS IIIA with 1.62 and MPS VI with 0.27 per 100,000 live births. The percentage of MPS II, MPS III, and MPS VI comprised 53%, 40%, and 7% of all cases. There was no case reported for MPS I, IV, and VII [47] (Table 2 and Table 3).

#### 2.8.15. Germany

In a retrospective epidemiological survey study in Germany between 1980 and 1995 (16 years), 474 cases were identified with MPS. During this period, 13,410,924 live births were reported by the German Bureau of Statistics. The combined birth prevalence of MPS was 3.51 per 100,000 live births. MPS III had the highest birth prevalence of 1.57 per 100,000 live births, comprising 44% of all cases. The birth prevalence of MPS I, II, IVA, and VI were 0.69, 0.64, 0.38, and 0.23, respectively, with percentages of 20%, 18%, 11%, and 7%. No case of MPS VII was reported in Germany. Additionally, this study showed that the number of births of Turkish origin was 617,013, which was a high proportion of patients with MPS IIIB (33%), MPS IVA (22%), and MPS VI (52%). However, these data accounted for less than 5% of total births [48] (Table 2 and Table 3).

#### 2.8.16. The Netherlands

The diagnosis of MPS was carried out between 1970 and 1996 (27 years), and a total of 331 MPS cases were reported during this period. The combined birth prevalence was estimated at 4.5 per 100,000 live births. MPS III (47%) had the highest birth prevalence of 1.89 per 100,000 live births, followed by MPS I (1.19 per 100,000 live births). The birth prevalences of MPS II, IVA, IVB, VI, and VII were 0.67, 0.22, 0.14, 0.15, and 0.24, respectively. The percentages of MPS I, II, IV, VI, and VII comprised 25%, 15.5%, 8%, 2%, and 2% of all MPS cases [21] (Table 2 and Table 3).

#### 2.8.17. Northern Ireland

In an epidemiological study of MPS conducted between 1958 and 1985, a total of 34 MPS cases and 839,517 live births were reported, 432,849 of whom were live male births. The combined birth prevalence was 4.0 per 100,000 live births. Among the MPS group, MPS I (41%) had the highest birth prevalence of 1.66 per 100,000 live births, followed by MPS IVA (32%) with 1.3 per 100,000 live births. The birth prevalence of MPS II (18%) and MPS III (9%) was 0.71 (1.39 live male births) and 0.36 per 100,000 live births. No case of MPS VI and MPS VII was reported [49] (Table 2 and Table 3).

#### 2.8.18. Poland

In a retrospective study to estimate the birth prevalence of MPS in Poland, 392 MPS cases were diagnosed, and the live births between 1970 and 2010 were reported as 21,686,890. MPS III had the highest birth prevalence of 0.86 per 100,000 live births, which comprised 48% of all cases. The birth prevalence of MPS I, II, IV, and VI was 0.22, 0.46, 0.14, and 0.0132 per 100,000 live births, which accounted for 12%, 25%, 8%, and 1%, respectively. No case for MPS VII was reported [50] (Table 2 and Table 3).

#### 2.8.19. Portugal

The diagnosis of MPS was carried out from 1982 to 2001. A total of 62 cases were diagnosed with MPS, with a combined birth prevalence of 4.8 per 100,000 live births. MPS I had the highest birth prevalence of 1.33 per 100,000 live births, accounting for 13%. The birth prevalence of MPS II, MPS III, MPS IVA, and MPS VI was 1.09 (34%), 0.84 (23%), 0.6 (16%), and 0.42 (10%) per 100,000 live births [19] (Table 2 and Table 3).

#### 2.8.20. The United States

The birth prevalence of MPS I was estimated depending on the newborn screening program for MPS I implemented in Illinois, Missouri, Kentucky, and Pennsylvania. The newborn screening is based on the measurement of IDUA enzyme activity in DBS.

In Illinois, 17,300 newborns were screened from November 2014 to December 2014, 17 of whom were called out, and the screening was repeated before reporting. The results showed that 15 newborns were false-positive, and two newborns were not confirmed [51].

In Missouri, 174,636 DBS were screened for MPS I, 70 of which were suspected. As a result, one patient had a severe form of MPS I, 58 newborns were false positives, nine newborns were not confirmed, and two samples were lost to follow-up. The birth prevalence of MPS I in Illinois and Missouri was 1.1 per 100,000 live births [51].

In the University of Washington study, Scott et al. evaluated MS/MS multiplex screening procedures for three LSDs using anonymous DBS from the Washington State NBS program. A cutoff of IDUA activity ≤ 1.15 µmol/h/L (corresponding to ≤32% of the mean) was used for MPS I. Therefore, the overall birth prevalence of infants (referred to as “may eventually develop clinical symptoms of MPS I”) was 1/35,700 (95% Cl: 1/43,000–1/11,100) [52]. The incidence of MPS in the United States was calculated by Puckett et al. for 10 years between 1995 and 2005. Data were obtained from the National MPS Society and the National U.S. Census Bureau. The combined birth prevalence of MPS was 1.2 per 100,000 live births. Within the MPS group, MPS III was the most common type. The birth prevalence of MPS I, II, III, IV, VI, and VII was 0.34, 0.29, 0.38, 0.09, 0.05, and 0.05 per 100,000 live births. The percentage of MPS I, II, III, IV, VI, and VII was 31.7%, 28.3%, 24.2%, 7.5%, 4.2%, and 4.2%, respectively [53] (Table 2 and Table 3).

## 3. Mutations

### 3.1. MPS I Mutations

MPS I results from a mutation in the IDUA gene located on chromosome 4p16.3. The prevalence of MPS I is the highest in the United Kingdom, the Netherlands, Germany, and Australia [1,54].

A total of 223 mutations were identified in the Human Gene Mutation Database, the highest proportion belonging to missense/nonsense mutations (130). Other mutation types in the IDUA gene are splicing (36), regulatory (1), small deletions (31), small insertions (16), small indels (1), gross deletion (4), gross insertions/duplications (1), and complex rearrangements (3) (http://www.hgmd.cf.ac.uk/ac/gene.php?gene=IDUA, accessed on 28 July 2020).

The worldwide IDUA mutations include p.W402X, p.Q70X, p.P533R, and p.G51D [1,55]. The distribution of pathogenic alleles shows variations among the populations.

p.W402X accounted for a significant portion of mutant alleles in the United States, with a frequency of 45%. Similar frequencies were also observed in Colombia, the United Kingdom, the Netherlands, and Germany, but were less frequent in other countries such as Norway, Italy, Austria, and Turkey [1,9,56]. In addition to this, the p.W402X pathogenic allele is only identified in Europeans (0.0014), Latinos (0.0004), and other countries with lower prevalence, while no cases are observed in Asian or African countries [9,55].

The second most frequent variant is p.Q70X, mostly recorded in Russia and Scandinavia with 50% frequency [57,58]. No cases have been reported in Asians and Latinos [9]. When comparing the p.Q70X mutation to the p.W402X mutation, the frequencies of both were relatively low in Italy and Turkey [59].

Another common variant observed in MPS I is the p.P533R allele, which has the highest frequency in Morocco, Tunisia, and Algeria, of which the frequencies were 92%, 81%, and 54%, respectively [60,61,62,63,64]. This variant was also recorded in the Mediterranean regions (42% of alleles) and North Europe [9].

Mutation p.G51D has the highest frequency in Norway with 54%. This variant is observed in Italy with 13.3%, Russia with 42%, Poland with 30%, and Austria with 31% of all alleles. However, Italy has shown different frequencies of mutations at similar rates, including p.W402X, p.P533R, and p.Q70X [9,65].

Some of the IDUA mutations are specific to Asian countries. c.704ins5 and p.R89Q mutations have been mostly reported in Japan, and the frequencies were 18% and 24%, respectively [66]. p.A79V and p.L346R mutations have been identified in China, the frequency of which accounted for 28.9% of all mutant alleles [67]. However, c.704ins5 and p.L346R mutations were also observed in Korea, with a frequency of 13.8% and 27.6%, respectively. A founder effect leads to the difference in these mutations in Japan, China, and South Korea [68].

There are recently published mutations specific for some nations, including Yemen (c.657delA) [69], Pakistan (p.L303P and c.1456G > T) [54,70], and Tunisia (c.1650 + 1G > T) [71]. Additionally, Iran has the novel widespread mutation p.Y109H, which accounts for 15.6% of identified mutations [72].

### 3.2. MPS II Mutations

A mutation in the IDS gene, which locates on chromosome Xq28, leads to MPS II. The prevalence of MPS II is the highest in Japan with 55%, South Korea at 54.6%, Taiwan with 52%, and China with 47.4% among all types of MPS [1,23,37,38].

Five hundred and fifty-two mutations in the IDS gene have been described involving 284 missense/nonsense, 51 splicing mutations, 102 small deletions, 45 small insertions, 11 small indels, 40 gross deletions, four gross insertions/duplications, and 15 complex rearrangements (http://www.hgmd.cf.ac.uk/ac/gene.php?gene=IDS, accessed on 28 July 2020).

Mutation p.R468 has a higher allele frequency that has been found most frequently in Asian countries. Codon R468 is adjacent to the positively charged active site residue, and the substrate enters into the active site cavity with the help of K347. Therefore, R468L or R468W, changing from the positively charged residue (arginine) to large hydrophobic residues (Leucine or Tryptophan), results in modification of the active site structure and also in a significant change in the substrate affinity [73]. From this point, codon R468 could be considered to be a mutational hotspot. In Taiwan, 42.8% of patients have MPS II. The p.R468 mutations contained p.R468W and p.R468Q mutations at an equal rate [74]. In the study of the Japanese population, 43 individuals identified with MPS II had p.468 mutations, including p.R468Q in three patients, p.R468W in one patient, and p.R468L in one patient [75]. Studies showed that IDS was highly heterogeneous in the mutation spectrum [73,76,77]. Among the Chinese population, IDS mutations showed variations. In a case report, three siblings had the same nonsense mutation, p.E344X. Many exonic point mutations were identified in China, and IDS mutations in Chinese patients were mostly located in exon 9, adjacent to exon 2 and against exon 3 [78]. IDS–IDS2 recombination mutations are quite common in some nations. According to research containing 49 Korean patients with MPS II, 30 patients described with the severe phenotype had the IDS–IDS2 recombination mutation. In this case, the IDS–IDS2 recombination mutations result in a severe phenotype [79].

Some common mutations are observed in the IDS gene that have spread across the world at a low frequency. R88 missense mutations were found in Europe and Russia; P86L, in Japan, Europe, and South Africa; c.1122C > T point mutation, in European countries, Russia and Korea; and W12X, in Italy [76,80];, c.1454T > A, c.162T > A, c.1327C > T, c.1019G > A, and c.1454T > A missense/nonsense mutations, in India [81].

We showed that the prevalence of MPS per 100,000 live births in 31 countries across the world in Figure 1. The incidence of MPS (%) was graphed to easily compare each MPS type (Figure 2). Moreover, the geographic distribution of combined birth prevalence of MPS was shown in Figure 3.

### 3.3. MPS III Mutations

#### 3.3.1. MPS IIIA

MPS IIIA arises from a mutation in the SGSH gene located on chromosome 17q25.3. The prevalence of this subtype is the highest in Northern Europe [82,83], and the carrier frequency of patients with MPS IIIA was estimated to be 1/7 to 1/10 [84].

A total of 142 mutations are listed for the SGSH gene in the HGMD: missense/nonsense mutations have 109 mutations, followed by 17 small deletions. Additionally, there are other mutation types in a limited number, including two splicing mutations, nine small insertions, three gross deletions, one small indel, and one gross insertion/duplication (http://www.hgmd.cf.ac.uk/ac/gene.php?gene=SGSH, accessed on 28 July 2020).

The most common SGSH mutation worldwide is R245H, followed by R74C and S66W [85]. The p.R245H mutation is most common in the British population, with a frequency of 20% [86]. However, a research study on 77 patients with MPS IIIA in the Cayman Islands showed that the R245H mutation was recorded in both patients and all carriers (46.75% of all suspected patients). This mutation was derived from a founder effect in the Cayman Islands population [84]. Moreover, this mutation has a high frequency of 57.8% in the Netherlands, Holland, with 56.7%, Germany with 35%, Poland with 56%, Australia with 31%, Spain with 45.5%, United Kingdom (19.2%), and Italy with 33% [82,83,87,88].

The R74C mutation accounts for nearly 50% of mutant alleles in Poland, while S66W comprises 33% of all alleles in Italy [83,87].

#### 3.3.2. MPS IIIB

A mutation in the NAGLU gene, located on chromosome 17q21, results in MPS IIIB disorder. The frequency of MPS IIIB is the highest in Greece, accounting for 81% of all MPS III cases and 45% of all MPS [89].

There are 156 mutations described in the NAGLU gene; the most significant proportion comprises 107 missense/nonsense mutations. Additionally, five splicing mutations, 23 small deletions, 13 small insertions, one small indel, four gross deletions, and three gross insertions/duplications have been found in the NAGLU gene (http://www.hgmd.cf.ac.uk/ac/gene.php?gene=NAGLU, accessed on 28 July 2020).

The most common mutations show variations among the geographic position of the countries. Y140C, H414R, and R626X are the most common mutations worldwide, particularly in Greece [89]. Specifically, Y140C was described by some of the countries as the most frequent mutation [90,91,92]. In a case report in Poland, a female patient diagnosed with MPS IIIB had a heterozygous mutation (c.638C > T/c.889C > T) in the NAGLU gene. Mutation c.638C > T (p.Pro213Leu) was transferred from father while c.889C > T (p.Arg297Ter), which is pathogenic, was taken from the mother. Both alleles were of wild types and in the heterozygotic configuration. This compound heterozygote (c.638C > T/c.889C > T) had a relatively high residual activity of α-N-acetylglucosaminidase, which was about 25%. This mutation has resulted in mild phenotypes arising from the partially preserved activity of the p.Pro213Leu variant of α-N-acetylglucosaminidase. Various cognitive and communication functions were preserved as well as somewhat better motoric functions at the age of 13 years [93]. Another study showed that c.1693C > T (p.R565W) and c.1914_1915insT (p.E639*) alleles in the NAGLU gene were the most common, at 66.6%. The other mutations observed in that study were c.1694G > T (p.R565L) and c.2209C > G (p.R737G) [77]. c.587C > T (p.Pro196Leu) was another novel mutation in the NAGLU gene, and the patient stemmed from a homozygote family. According to the result of this study, p.Pro196Leu has a toxic effect on protein structure and function [94]. In Tunisia, 10 patients from six different families were diagnosed with MPS III, and three of them were diagnosed with MPS IIIB. These three patients had a missense (p.L550P) and nonsense (p.E153X) mutation in the NAGLU gene. p.L550P allele was found in two homozygote siblings who came from heterozygote parents. In contrast, p.E153X was recorded in homozygosity in the third child who came from a heterozygote parent [95]. The other study conducted in Egypt revealed that Y558*, L550, R297*, R482W, G79C, Y140C, and W268R mutations were common, as well as three novel homozygous mutations. Homozygous mutation (8/10) comprises 80%, while one compound heterozygous mutation (Y140C/W268R) accounted for 10%, and one heterozygous mutation consisted of 10% of all mutations. This was the first study performed on MPS IIIB patients in Egypt [91].

Similarly, there are some novel mutations reported by other countries and researchers, such as c.457G > A (p.E153K) from Iran [96]; compound heterozygous genotype p.Y140C/p.R297X from Canada [97]; c.1705C > A (p.Q569K) novel and c.1562C > T/c.1705C > A heterozygote mutations from China [98]; and K255Rfs*18, E153Rfs*39, and Q350* from Egypt [91].

#### 3.3.3. MPS IIIC

Mutations in the HGSNAT gene, located on chromosome 8p11.1, cause MPS IIIC. The prevalence of MPS IIIC per 100,000 live newborns is highest in the Netherlands (0.21), Portugal (0.12), and Australia (0.07) [19,20,21].

Sixty-six mutations have been found in the HGSNAT gene involving 37 missense/nonsense, 14 splicings, five small deletions, five small insertions, one small indel, two gross deletions, one gross insertion/duplication, and one complex rearrangement (http://www.hgmd.cf.ac.uk/ac/gene.php?gene=HGSNAT, accessed on 28 July 2020).

A research study in Tunisia presented that splice site mutation c.234 + 1G > A (IVS2 + 1G > A) was frequent among MPS IIIC patients in Mediterranean countries. Two unrelated Tunisian patients in this study also carried mutation c.234 + 1G > A in the HGSNAT gene, which suggested that the origin of these patients was based on Mediterranean countries [95]. Previously reported data illustrated that this mutation had a higher frequency in patients with MPS IIIC from Spain, Turkey, France, Italy, and Morocco [99,100]. c.1360C > T was another mutation with high frequency in Colombia. This mutation leads to a change from glutamine to early termination at codon 454, and it was recorded in the homozygous state in all five patients (total of 11 affected patients) [101]. In a Korean study, there were two mutations containing c.234 + 1G > A (IVS2 + 1G > A) and c.1150C > T (p.Arg138*) in a female patient. The patient had a compound heterozygous mutation c.234 + 1G > A/c.1150C > T [102]. The c.1150C > T nonsense mutation in the HGSNAT gene was observed in Poland, Czech Republic, Italy, the Netherlands, Canada, and Turkey, indicating that this mutation has relatively high frequency among families with MPS IIIC. However, it has not been established as a founder effect of MPS IIIC in any nation, since this mutation has a broad range of distribution [99,100,102,103,104].

There are various mutations in the HGSNAT gene depending on the geographical regions; c.493 + 1G > A and c.1030C > T in Singapore; c.744 − 2A > G and c.887C > A in Pakistan; c.852 − 1G>A and c.848C > T in Turkey; c.1250+1G > A in Turkey, the United Kingdom, and the United States; c.1542+4dupA, c.1209G>T, c.1441G > T, c.1843G > A, and c.1674C > G in the United Kingdom; c.1-1925_118+296del in Turkey; c.641delG in Belarus; c.739delA in North Africa; c.1271dupG, c.1622C > T, and c.1411G > A in Greece; c.1516C > T in Belgium; c.410T > C, c.1457G > A, c.1553C > T, and c.1030C > T in Germany; and c.1466C > A in the United States [99].

#### 3.3.4. MPS IIID

MPS IIID is caused by a mutation in the GNS gene locating on chromosome 12q14.

A total of 23 mutations have been identified in the GNS gene, which involves seven missense/nonsense, three splicings, four small insertions, four small deletions, one small indel, two gross deletions, and two complex rearrangements (http://www.hgmd.cf.ac.uk/ac/gene.php?gene=GNS, accessed on 29 July 2020).

MPS IIID is considered rare compared to other MPS types, and thus there are limited data about its mutations. Since 1980, when MPS IIID was identified, only 23 disease-causing mutations in the GNS gene have been described. A study recorded 15 novel mutations in 16 patients with MPS IIID, which contained missense mutations c.281G>T (p.S94I), c.1019A > G (p.K340R) and c.1253G > A (p.G418E), nonsense mutation c.1162A > T (p.K388X), splice site mutations c.875+2delT, c.1097_1098+1delAGGinsGGT, c.1309 − 2A> G and c.1420 − 2A > G, frameshift mutations c.59_66del8, c.109dupG and c.1231dupG, large deletions Del EX1+, Del EX6, 7 (c.625-637_875 + 6del3346ins8) and delEX9-14 (x.1046_1659 + 16210del36529ins9), and in-frame small deletion c.911_919del9 (p.E304_L306del). In this study, all nonsense mutations, insertions, and deletions leading to frame-shifts were considered pathogenic, which has a detrimental effect on protein structure and outcomes [105]. In another study conducted by Italian researchers, a family over four generations was examined, and the results revealed two different MPS types, including MPS IIID and MPS IV. In other words, this family was the genetic carrier of both MPS IIID and MPS IV. The splice site mutation, c.1098+1G>A, was found in the GNS gene in the parent and their child. The constellation of the clinical malformation and skeletal abnormalities were inconsistent with previously reported in MPS IIID patients [106]. An in vitro study was performed to detect the mutation spectrum of the GNS gene, the ramifications of which found a homozygous nonsense mutation R355X (1063C→T) [107]. Four patients with MPS IIID, two of them siblings and the other two from unrelated families, were carrying three different mutations in the GNS gene. Both siblings had homozygous nonsense mutation c.1168C > T. However, the other patient was heterozygous for a splice recognition site mutation c.876 − 2A > G, whereas the final patient coming from third family carried a homozygous frameshift mutation c.1138_1139insGTCCT [108].

### 3.4. MPS IV Mutations

#### 3.4.1. MPS IVA

MPS IVA arises from a mutation in the GALNS gene, which locates on chromosome 16q24.3.

There are 327 mutations found in the GALNS gene; the most significant number (242) belongs to missense/nonsense mutations. This gene contains 32 splicing mutations, 32 small deletions, five small insertions, two small indels, nine gross deletions, two gross insertions/duplications, and three complex rearrangements (http://www.hgmd.cf.ac.uk/ac/gene.php?gene=GALNS, accessed on 29 July 2020).

MPS IVA has a wide range of allelic heterogeneity, and thus its mutation spectrum is quite extensive. In India, the most common mutations in the GALNS gene were the missense mutations p.Ser287Leu at 8.82%, p.Phe216Ser at 7.35%, p.Asn32Thr at 7.24%, and pAla291Ser at 5.88%. Mutant alleles in exon 1, 7, and 8 were 45.65% of the mutations [109]. Furthermore, another study illustrated that there had been 148 unique mutations in the GALNS gene, 26 of which were novel mutations. As shown in other studies, missense mutations had the biggest proportion, with 78.4% of the mutant alleles analyzed in that study. The most frequent mutations were p.R386C, p.G301C, and p.I113F missense mutations, accounting for over 5% of all mutations [110]. The results obtained from the study in Brazil indicated that 68 unrelated South-American patients with MPS IVA had 25 different mutations, seven of which were novel involving c.-1_6delinsT, p.Asp45Gly, p.Asn76Lys, c.319 + 2T > C, p.Ser120Leu, c.759 − 2A > G, and p.His236Arg. The most common mutations in Brazil were p.Ser341Arg (22%), p.Gly301Cys (13.4%), p.Arg386Cys (12.6%), p.Arg94Leu (11%), and p.Gly116Ser (8.6%). The p.Ser341Arg is the most prevalent mutation in the GALNS gene. Similarly, these alleles have been frequently reported across the world. All patients with the p.Ser341Arg mutation had the same haplotype. This information implied a founder effect, derived from the same ancestor [111]. p.I113F and p.T312S alleles were described only in British/Irish patients, with the frequency of 18% and 14% of all mutations [112,113]. PM1 and p.W10X were the most common mutations among patients with MPS IVA in Italy, comprising 26.7% and 13.3% of all mutant alleles studied [114]. c.334delG and c.708delT deletions accounted for 40% and 20% of mutant alleles among the patients with MPS IV in Turkey [115]. Some of the frequent mutations are c.1156C > T, c.337A > T, c.901G > T, c.120 + 1G > A, c.935C > G, c.871G > A, c.860C > T, c.953T > G, and c.757C > T [116]. p.Arg386Cys for Spanish, Argentines, Chinese, Italian, Colombian, Polish, Turkish, and Chilean; p.Gly301Cys for Colombian, Portuguese, and Spanish; c.120 + 1G > A for Tunisian; p.Met391Val for French Canadian, French, Canadian Caucasian, American Caucasian, and German; p.Ala291Thr for Asian-multiethnic, British, Finnish, Pakistani, Chinese, and Japanese; p.Ser287Leu for Middle Eastern, Turkish, Spanish, Polish, Greek, Macedonian, Austrian, New Zealander, and Irish/Italian/Polish; p.Met318Arg for Chinese, Taiwanese, South-East Asian, and Japanese; and p.Arg235Trp for Pakistani have been mostly described by researchers [116].

Overall, there has been a broad range of mutational spectrum in GALNS, and the distribution of the alleles varies from one geographic region to another. However, some of the mutations have still been recorded with the highest rates. The founder effect is important to reveal the origin of these mutations. For that purpose, both mutational and haplotype analyses have been carried out together.

#### 3.4.2. MPS IVB

A mutation in the GLB1 gene located on chromosome 3p21.33 leads to MPS IVB.

Two hundred and nine mutations have been described in the GLB1 gene, 159 of which comprise missense/nonsense mutations. Additionally, 16 splicing mutations, 17 small deletions, 12 small insertions, two small indels, one gross deletion, and two gross insertions/duplications were also found in this gene (http://www.hgmd.cf.ac.uk/ac/gene.php?gene=GLB1, accessed on 29 July 2020).

The most important thing to be noticed is that mutations in the GLB1 gene cause MPS IVB and GM1 ganglioside. That is to say, the deficiency of the ß-galactosidase enzyme results in the accumulation of ganglioside GM1 and keratan sulfate. If the accumulation of keratan sulfate predominates, patients are diagnosed with MPS IVB [1]. The most common mutations in the GLB1 gene related to MPS IVB are p.Y183H, p.W273L, and p.T500A [117,118,119]. Notably, the p.W273L mutation was found commonly among European MPS IVB patients who were unrelated, of which the frequency was 79% of all alleles [120]. In a mutational study among Gypsies, there have been five patients with MPS IVB. Out of five, two patients had the p.T500A mutation, and four patients had novel mutations p.R401H, p.D441N, and p.Y83C. Moreover, one of the patients had a heterozygous genotype (p.D441/p.Y83C), while another patient was homozygous for the p.R401H mutation [121]. A Macedonian girl who was 24 years old was described for MPS IVB. The results of the genetic analysis demonstrated that the girl carried the p.H281Y allele, which was a common disease-causing mutation and had higher allele frequency among Caucasian patients, and the p.W273R novel mutation [122]. According to a report, out of 51 cases, 41 cases were described for MPS IVB. The same study indicated 28 GLB1 variants in patients included in this study. The most frequent variant was W273L, which was consistently related with pure MPS IVB, followed by T500A. This report showed that pure MPS IVB was also observed in a patient carrying R201H homozygous variant and in the majority of the T500A variant. Furthermore, Y333C, G438E, T82M, R201H, and H281Y variants were also reported in the same study [119].

### 3.5. MPS VI Mutations

MPS VI is caused by a mutation in the ARSB (or 4S) gene located on chromosome 5q11–q13.

The ARSB gene comprises 192 mutations, including 143 missense/nonsense mutations, 11 splicing, 24 small deletions, six small insertions, two small indels, and six gross deletions (http://www.hgmd.cf.ac.uk/ac/gene.php?gene=ARSB, accessed on 29 July 2020).

Between 1983 and 2016, a total of 18 cases from six unrelated consanguineous families were confirmed with the severe phenotype. Two different genotypes were detected in the ARSB gene; the homozygous nonsense mutation c.753C > G (p.Y251X) in all of five families in Al-Hofuf and a novel homozygous frameshift mutation c.270_274del5bp pc.91Afs*34 in a single-family in Abha [123]. To detect the ARSB gene mutation and clinical presentation of MPS VI in Iran, suspected MPS VI samples were collected for eight years from 2010 to 2018. Of 14 patients from 10 unrelated consanguineous families, 13 originated from Arab ethnicity with the severe phenotype and one from Fars ethnicity with the mild phenotype were diagnosed for MPS VI. Molecular analysis demonstrated that the ARSB gene had four pathogenic homozygous missense and nonsense mutations involving one novel nonsense mutation c.281C > A (p.Ser94X) in nine patients, one previously reported nonsense mutation c.735C > G (p.Try251X) in three patients, and two missense mutations c.904G > A (p.Gly302Arg) and c.454C > T (p.Arg152Trp) in two patients. From the results of mutational analysis, it is evident that c.281C > A (p.Ser94X) is the most frequent among Iranian Arab patients with MPS VI [124]. In Turkey, out of 21 patients with MPS VI, some received ERT for at least six months, while others could not; 17 patients (81%) were diagnosed with the severe form, while the mild form had only four patients (19%). The most common mutation in the ARSB gene was p.L321P, which comprises 58.8% of all alleles. This showed that there was a founder effect in patients with MPS VI in the Turkish population. However, two novel mutations have been determined: p.G79E and p.E390K. In addition, p.L321P, p.C192R, p.E346fs*13, p.R160*, and p.R191 mutations were reported. Furthermore, there were two homozygous polymorphisms: V358M and IVS5-28AáC. p.G79E and p.E390K novel missense mutations were reported in the only mild forms of MPS VI. In contrast, p.R160*, p.R191*, p.L321P, p.E346Sfs*13, and p.E390K mutations were recorded as the only severe forms of the disease [125]. A study in Taiwan showed that the ARSB gene carried eight polymorphisms in homozygous alleles in the patient and her family members, containing c.246G/A (p.L82L), c.342C/T (p.I114I), c.370C/T (L124L), c.972A/G (G324G), c.1072G/A (p.V358M), c.1126G/A (p.V376M), c.1191A/G (p.P397P), and c.1515C/T (p.Y505Y). c.1072G/A (p.V358M) and c.1191A/G (p.P397P) polymorphisms were detected earlier in MPS VI patients in Taiwan. Although the prevalence of MPS VI is lower (three patients diagnosed with MPS VI in 11 years; one in 875,000 newborns), the variety of mutations is much greater in Taiwan [126,127,128,129]. Research conducted in Brazil indicated that the p.H178L missense mutation had a founder effect on Brazilian MPS VI patients. During the study period, 236 unaffected blood samples were collected from patients with MPS VI and their relatives. Nine living and four deceased individuals in 11 consanguinities were diagnosed with MPS VI. A total of 98 (20.8%) mutant alleles and 374 (79.2%) normal alleles were described, with 41.5% of the individuals heterozygous for p.H178L mutation and 58.5% homozygous for the normal allele [130].

Some of the novel ARSB mutations were as follows: c.870G > A (p.Trp290stop) [131]; c.716A > G [132]; c.1213 + 5G > T (IVS6 + 5G > T) [133]; p.H178, p.H242R, p.*534W, and IVS5 + 2T > C [134].

### 3.6. MPS VII Mutations

MPS VII is rooted in a mutation in the GUSB gene located on chromosome 7q21.11.

There have been 63 mutations in the GUSB gene, including 52 missense/nonsense mutations, five splicing mutations, one regulatory, four small deletions, and one gross deletion (http://www.hgmd.cf.ac.uk/ac/gene.php?gene=GUSB, accessed on 29 July 2020). A case report revealed a novel mutation (c.542G > T, p.Arg181Leu) in the GUSB gene of an Iranian female patient (28 months old) with MPS VII. The parent of the affected individual was heterozygous for this variant [135]. Another research study indicated that a male patient at the age of 22.5 years, diagnosed with MPS VII, had two missense mutations (p.R382H (c.1145G > A) and p.Y508C (c.1523A > G)) in the GUSB gene [136]. c.211_214del (p.S72Afs*34) and c.1270C > T (p.H424Y) were the novel compound heterozygous mutations found in a patient with MPS VII in China. As shown above, missense mutations have the highest percentage of all types of mutations in the GUSB gene. This is due to the fact that missing TCAG sequences are located at the beginning of exon 2, and the mutation may break the splicing site. The data analysis in China indicated that the c.211_214delTCAG mutation results in a frameshift and affects the splicing of mRNA [137]. A new case report from Italy demonstrated that the patient with a severe form carried the homozygous variant c.1617C > T (p.Ser539=), which leads to aberrant splicing with partial skipping of exon 10 and complete skipping of exon 9 [138]. Another report illustrated a homozygous mutation p.N379D (c.1135A > G) (Asn379Asp) in the GUSB gene of an affected fetus and in a heterozygous in both parents. This substitution might be deleterious to the protein structure or even function, because Asn379 is conserved in the GUSB gene [139]. Although MPS VII is a rarer genetic disease, there have been various mutations in the GUSB gene. The most frequent missense mutations were as follows: c.526C > T (p.L176F) with 20.4% across the world; c.1244C > T (p.P415L) with 4.9%, c.1222C > T (p.P408S) with 4.9%, and c.1856C > T (p.A619V) with 4.9% for the Mexican and Japanese population; c.646C > T (p.R216W) with 3.9%, c.1144C > T (p.R382C) with 3.9%, and c.1429C > T (p.R477W) with 3.9%. p.R357X nonsense mutation was the second most common mutation in the GUSB gene [140,141,142,143,144,145,146,147,148].

### 3.7. MPS IX Mutations

MPS IX arises from a mutation in the HYAL1 gene located on chromosome 3p21.3–p21.2. HYAL1 carries three mutations containing one missense/nonsense mutation, one small deletion, and one complex rearrangement (http://www.hgmd.cf.ac.uk/ac/gene.php?gene=HYAL1, accessed on 29 July 2020).

Until now, only four patients with MPS IX have been described in the literature [149,150]. Two mutations have been identified in the HYAL1 gene of a patient with MPS IX, which were c.G1412A (p.Glu268Lys) inherited from the father, and 1361del37ins14 inherited from the maternal side of the family. c.G1412A is a non-conservative amino acid substitution in a putative active site residue, and 1361del37ins14 is a complex intragenic rearrangement resulting in a frameshift and a premature termination codon on one allele. According to the authors in that study, both 1361del37ins14 and 1412G→A mutations destroy the activity of HYAL1 [151]. Another Middle Eastern study involved three siblings diagnosed with MPS IX. Molecular analysis of the hyaluronidase genes of three affected siblings revealed a homozygous deletion of c.104delT, resulting in a premature termination codon p.Val35AlafsX25 in the HYAL1 gene of these three siblings. The parents of these siblings were carriers of this deletion with the heterozygosity. However, there was no homozygosity among the other nine family members included in the study. Moreover, there was consanguinity between the father and mother of these siblings [149].

The results indicate that MPS IX is a rare disorder that lacks a clear diagnosis method except for juvenile idiopathic arthritis [152].

## 4. Frequency of MPS with Newborn Screening

Newborn screening (NBS) is a screening program that has become available for MPS. NBS has been able to identify conditions affecting a child’s long-term health or survival. Early diagnosis and treatment can prevent death and disabilities [153]. At first, NBS was used for phenylketonuria with bacterial inhibition assay to quantify phenylalanine levels in DBS in 1963 [154,155]. Since then, NBS has gained speed in diagnosing LSDs, and it has started to become utilized for Krabbe, Pompe, Fabry, Gaucher disease, MPS I-II, and Niemann–Pick A/B [156].

In recent years, many technological methods have improved NBS, including radioimmunoassay, tandem mass spectrometry (MS/MS), calorimetric and fluorometric immunoassays, isoelectric focusing, high-performance liquid chromatography (LC-MS/MS), and molecular testing [155]. However, among all methods, MS/MS technologies have offered efficient solutions in the analysis of proteins and substrates on DBS when used together with NBS [157,158]. NBS, as a first-tier screening test, has been firstly approved in the USA for MPS I [156]. In 2016, MPS I was officially added to the recommended uniform screening panel in the USA, which enabled MPS I to be universally screened [159,160,161].

After the successive implication of MPS I, the NBS of MPS II has universally begun in the state of Illinois (USA) (Table 4). Iduronate-2-sulfatase (I2S) activity was measured in DBS by using LC-MS/MS. In this study, 93,219 infants were screened, and nine infants were suspected of low enzyme activity of I2S. Of nine infants, one had low I2S activity in plasma (3.13 nmol/4 h/mL; normal level; 155–1082 nmol/4 h/mL), elevated uGAGs (128.2 mg/mmol creatinine; normal level; 0–53 mg/mmol creatinine), and a common pathogenic variant p. R468Q in the I2S gene. For this patient, ERT started at three weeks of age. Five infants had pseudodeficiency for I2S, normal uGAG level, and a novel mutation in the I2S gene. Three infants were found to have normal plasma I2S activity. This finding showed that NBS for MPS II was effective and could be implemented with a low rate of false-positive results [179]. Nearly 100,000 DBS had been tested with LC-MS/MS in the Washington State NBS Laboratory for MPS II, MPS IIIB, MPS IVA, MPS VI, and MPS VII (Table 4). The samples with enzyme activity less than 10% of the daily mean were isolated, and the relevant genes were investigated with the DNA sequence analysis. Five samples had more than one screen positive result. For MPS II, the mean enzyme activity of I2S was detected as 16.52 µmol/h/L blood in 105,214 samples. The enzyme activity of 18 samples was below 10%, seven of which had activity less than 5% of the daily mean. The other important point is that 11 samples, the enzyme activity of which was between 5% and 10% of the daily mean, did not have a pathogenic genotype. Seven samples had enzyme activity less than 5% of the daily mean, one of which was affected, and two were at risk of developing clinical symptoms. Approximately 53,000 male samples were included in this study due to the fact that MPS II is X-linked and male samples are more important in terms of high frequency. NBS result showed that the clinical frequency of MPS II was between 1/53,000 and 1/18,000 male samples. For MPS IIIB, 103,001 samples were screened, and the mean enzyme activity was 2.92 µmol/h/L blood. No less than 10% enzyme activity was observed in any samples. For MPS IVA, 106,106 samples were screened; the mean enzyme activity was 2.10 µmol/h/L blood. The enzyme activity of only eight samples was below 10% of the daily mean. One sample was carrying pathogenic variant p.F167V with heterozygosity, and one sample was homozygous for pA231G. For the 103,259 MPS VI samples that were screened, the mean enzyme activity was 14.63 µmol/h/L blood. The enzyme activity of four samples was below 10% of the daily mean, one of which was even less than 5% of the daily mean (0.32 µmol/h/L blood-enzymatic activity) and three of which were above 5% of the daily mean, and they were thought to have common pathogenic nucleotide variants, p.V358M and p.V376M. In terms of MPS VII, 94,931 samples were screened, and the enzyme activity was found to be 43.10 µmol/h/L blood. Enzymatic activity of one sample was less than 5% of daily mean that was 0.11 µmol/h/L blood, and this sample had two pathogenic variants, p.P67L and p.A442V, in the related gene. The results of this study have shown that LC-MS/MS could detect and quantify the enzyme activity on DBS [181].

In Taiwan, Lin et al. [170] also reported 35,285 newborns samples for MPS I in 2013, which were screened by fluorometric enzyme assay (Table 4). The 58 samples had less enzyme activity than the reference range (9.03–69.52 µmol/L blood*20 h), and those patients gave the second DBS samples. At the final stage, only three patients still had lower enzyme activities than the reference value, and the frequency of MPS I in this study was found to be 1/17,643. For three patients, uGAG quantification, uGAG two-dimensional electrophoresis (2-D EP), leukocyte enzymatic assay, and molecular DNA analysis were carried out. The results of this study showed that IDUA activity was deficient in two newborns (6.8–37.0 µmol/g protein/h), and the other newborn was thought to be a carrier depending on the reduced IDUA activity (3.5 µmol/g protein/h). uGAG quantification results were in the normal range for all three patients (reference range; 44.6 ± 23.7 mg/mmol creatinine); however, the 2-D EP results showed unclear HS but significant DS level for two affected patients. The third patient (probably carrier) was normal. Finally, DNA analysis confirmed the mutations in the two newborns. One patient had two missense mutations; c.303G > A (R105Q) variant inherited from the father and c.484G > A (R105Q) variant inherited from the mother. The second patient had the point mutation c.344G > T (D119Y) inherited from the father and c.99T > G (H33Q) inherited from the mother. Carrier suspected newborns and newborns’ parents were also confirmed to have MPS I carrier by DNA analysis. These results indicated that NBS with fluorescence enzyme assay showed high efficiency to detect IDUA enzyme activity on DBS [170].

In 2019, in Taiwan, Chan et al. [173] reported another study of NBS using LC-MS/MS. NBS has been routinely implicated for MPS I, II, and VI (Table 4). More than 100,000 DBS samples were screened, and the enzyme activities on DBS were measured by LC-MS/MS. Genotyping was done for some samples, and enzyme activity was under cutoff value. For MPS I, 130,237 newborns were screened, and 120 (0.009%) had lower enzyme activities than the cutoff value (<5% of mean activity). Second DBS samples were collected from those under cutoff value, and the only five samples (0.004%) showed reduced enzyme activity on both first and second DBS samples. After genetic analysis of five samples, compound heterozygous mutations (c.300-3C > G, c.1037T > G, c.1079T > G, c.1091C > T, c.1874A > C, c.1359_1384del) were detected in the IDUA gene. These results revealed that these five patients have attenuated MPS I [173].

As of MPS II, there were two different periods when samples were collected. From August to December 2015, 28,799 samples were screened, and 56 newborns were recalled for second DBS samples, 53 of which had low enzyme activities. Genetic sequence analysis was applied to these samples. As a result, c.301C > T (benign polymorphism; 16 samples with 11–25% of mean enzyme activity), c.1499C > T (non-pathogenic; 18 samples with normal range enzyme activity), linked variants c.103 + 34_56dup and c.851C > T (uGAGs in normal range but lower leukocytes enzyme activities; 3.2 nmol/Hr/mg protein, both in 16 samples), and c.890G > A (1 sample) mutations were detected in IDS gene. Between January 2016 and August 2017, 101,376 newborns were screened, and 184 gave second DBS samples. As a result, 96 samples with lower enzyme activities than 10% of the mean were applied to further genetic sequence analysis. Successively, 50 newborns had the c.1499C > T variant, and 38 newborns had lower leukocyte enzyme activity than the cutoff value (uGAG and physical examination were normal) linked with variants c.103+34_56dup and c.851C > T (pseudodeficiency). Furthermore, six uncertain significant variants c.589C > T (1 case), c.890G > A (1 case), and c.1478G > A (4 cases), and three asymptomatic MPS II cases (c.311A > T, c.817C > T and c.1025A > G) were detected. The enzyme activity of newborns who carried c.311A > T, c.817C > T, and c.1025A > G variants was significantly low (<3% of mean) in DBS and leukocytes. At the same time, HS (7.4–21.2 µg/mL) and DS (1.8–103.4 µg/mL) in urine were higher than the reference cutoff (<0.8 µg/mL for HS and <0.41 µg/mL for DS), which was the proof of the identification of asymptomatic MPS II in these patients [173].

For MPS VI, 131,075 newborns DBS samples were screened, 176 (0.13%) of which gave the second DBS. Only two samples had lower enzyme activity than 30% of mean activity in DBS samples (17.8 and 11.8 µmol/L/h) without any variations in the ARSB gene. However, the second sample (18.5 and 12.0 µmol/L/h) had one novel variant c.716A > G, respectively. These two patients had a normal range of leukocyte enzyme activity and uGAG-disaccharides, and no MPS VI patient was detected in this study. Therefore, a large group of newborns, the enzyme activity of which was below the cutoff value, were considered to be false positives at the beginning (especially for MPS II samples). This showed that the cutoff value was high and that uncertain significant variants highly affected these populations. Genetic sequence analysis as a second-tier test was confirmed with correct results. That is why the second-tier test is critical in the NBS program [173].

In an NBS study in Taiwan for Morquio disease and other LSDs, 73,743 newborns were screened by the eight-plex assay from 2018 to 2019 (Table 4). Six newborns showed low GALNS enzyme activity and biallelic GALNS variants, which made the incidence of 1/12,291 newborns. Of six newborns, one had 0% GALNS activity, and thus the incidence of MPS IVA was calculated as 1/73,743 (1 in 13,020 to 417,750), which was in line with the previous data in Taiwan (1/300,000 live births). Moreover, one case for MPS I, three cases for MPS II, three cases for MPS IIIB, and nine cases for Pompe, Gaucher, and Fabry disease were identified, making the incidence of 1/3206 newborns for LSDs [174].

Lin et al. (2019) studied total uGAG for a high-risk screening of MPS. The study included 153 clinically suspected children for MPS during the five-year period between 2013 and 2018. Forty urine samples were collected, and total GAG was analyzed by dimethylmethylene blue (DMB) spectrophotometric method. Two-dimensional electrophoresis was performed to evaluate the GAG disaccharide pattern, and LC-MS/MS was applied for the urine quantification of HS, DS, and KS. According to the results of these all methods, 22 patients with elevated uGAG levels, abnormal GAG-derived disaccharide pattern, and elevated urinary DS, HS, and KS levels gave blood samples to determine the enzyme activity and proteins in the leukocytes, which were measured by 4-methylumbelliferyl method and Coomassie Plus protein assay, respectively. Furthermore, genetic testing and counseling were undertaken for 13 patients who had an enzyme activity under 5% of the average value of the normal population. Therefore, three patients with MPS I, four patients with MPS II, five patients with MPS IIIB, and one patient with MPS IVA were diagnosed, 77% of whom were under three years of age. The most crucial point of the study was that no false-negative result was recorded in the quantification of urinary DS, HS, and KS, compared to the DBS studies in the literature [181]. On the other hand, the measurement of total uGAG at a high-risk population could lead to missing some patients suffering from especially MPS IV, VII, and IX.

In 2017, Bravo et al. [167] reported the results of NBS in Brazil by fluorometric enzyme assays of LSDs. In this study, 10,567 newborns were screened, and four babies were found to be suspected of LSDs (two for MPS I, one for Pompe disease, and one for Gaucher disease) (Table 4). Enzyme activities of these samples were measured in DBS, plasma, and leukocytes by fluorometric enzyme assay. uGAG analysis was conducted by the DMB assay and monodimensional electrophoresis. One of the MPS I babies showed lower enzyme activity (0.8 µmol/L/h) than the reference range (>5.0) on DBS. uGAG analysis of this baby was a normal level, and IDUA activity in DBS was undetectable compared to plasma in which IDUA activity was a normal level, and leukocytes enzyme activity was below (11 nmol/h/mg protein) the reference range (27–171 nmol/h/mg protein). The molecular analysis of the IDUA gene showed that this baby had two variants: c.251G > C (p.Gly84Ala), which was thought most likely to be pathogenic, and c.246C > G (p.His82Gln), which was benign and probably related to pseudodeficiency. All results together for the first baby indicated that this baby had pseudodeficiency for MPS I. The second suspected baby was detected to have low IDUA enzyme activity (2.4 µmol/L/h), normal uGAGs in the quantitative and qualitative analyses, and low enzyme activity measured in leukocytes (27 nmol/h/mg protein). Genetic sequence analysis of the IDUA gene revealed that the second baby had a heterozygous pathogenic variant c.1205G > A (p.Trp402Ter) inherited from the father. The results implied that this baby was an MPS I carrier. This study proved the importance of second-tier testing for the NBS program [167].

In another study on 20,018 DBS samples for LSDs in Mexico (Table 4), two MPS I patients were detected, which had compound heterozygous mutations: c.965T > A (p.Val322Glu) including previously reported c.1861C > T (p.Arg621Ter) and a variant of unknown significance (VUS) c.701G > C (p.Ser234Thr). The incidence rate of MPS I was calculated as 1/10,009 newborns, which was compatible with previous reports: 1/17,643 in Taiwan and 1/10,750 in Washington, USA [161]. Higher incidence rates of MPS detected by NBS proves that NBS could provide opportunities to identify MPS not recognized by traditional screening methods if supported by second-tier tests. It is clear that NBS, together with confirmatory tests, reveals pseudodeficiency alleles causing false positives and variants of unknown significance.

The results of NBS for LSDs by LC-MS/MS in North-East Italy showed the incidence rate was 1/4441 newborns (Table 4). A total of 44,411 DBS were collected in that study, 40 (0.09%) of which had lower enzyme activity than 0.2 multiple of the median. After confirmatory testing, 20 (50%) neonates were confirmed to have low enzyme activity, 10 of whom were affected by Pompe disease (two cases), Gaucher disease (two cases), Fabry disease (five cases), and MPS I (one case); however, the other 10 newborns had previously reported pseudodeficiency alleles with a high incidence rate. The most common pseudodeficiency allele observed in this study was p.Ala79Thr, which the newborns coming from African descents mostly carry. For the MPS I case, homozygous p.Pro553.Arg mutation was determined [169].

Additionally, screening studies of MPS and other disease types in other countries have been done to improve the efficiency of NBS. The multiplex of MS/MS assays was performed for MPS II, MPS IIIB, MPS IVA, MPS VI, MPS VII, and fucosidosis [182].

## 5. Discussion

We have summarized “epidemiology of mucopolysaccharidoses” in this review article by collecting 189 publications related to the prevalence of MPS by PubMed as of December 2020. We have demonstrated (1) that the prevalence of MPS varies in each country and ethnic background, (2) that the frequency of each type of MPS is affected by the founder effect, and (3) that NBS contributes to the accurate incidence of MPS.

As mentioned above, two-tier NBS detects more affected cases and provides an accurate incidence for MPS (including unrecognized late-onset types) in various geographic regions or different ethnic backgrounds.

Until now, a wide range of methods to measure GAGs and enzyme activity (the deficient enzymes shown in Table 1) have been developed. The pathogenicity of new mutations identified in the asymptomatic newborn screening positive neonates cannot be defined only by the molecular analysis; however, the recent reports suggest the clinical severity can be predicted by assaying specific GAGs as a second-tier screening. Therefore, it is critical to measure both enzyme and primary stored GAGs to define the phenotype (pseudodeficiency, attenuated, or severe) along with the molecular analysis [157,183,184]. In general, traditional methods cannot identify all patients before the patient becomes symptomatic; however, we need early diagnosis and early treatment for MPS patients since some symptoms are irreversible or hard to improve. Early interventions will prevent most complications.

Most LSDs could benefit from an early diagnosis since the availability of treatments leads to better consequences when starting at an early stage [184,185,186]. ERT and HSCT are clinically used for some MPS types, while gene therapy, SRT, and pharmacologic chaperons are under development. Studies across the world have shown that the efficacy of ERT and HSCT depends on early diagnosis of the disease [18,128,187,188,189].

Table 2 shows the prevalence of MPS in 27 countries, in which most patients were identified by traditional methods. The prevalence of several types of MPS has increased in several countries after the NBS application (Table 4), compared to the prevalence calculated by the traditional screening methods. NBS method has revealed carriers, pseudodeficiency, and true positives, supported by second-tier methods. Moreover, several pilot studies have shown similar incidences to traditional studies because of a limited number of newborns screened. To define the exact incidence of MPS, we need to perform newborn screening with more newborns in multiple countries and regions for a long term.

Understanding of the precise incidence of each type of MPS (Table 3) contributes to the understanding of natural history for MPS, etiology of the disease, specificity and sensitivity of screening system, frequency of clinical phenotype (attenuated or severe), carrier detection, follow up and life expectancy, and marketing of the drug development.

Therefore, it is worthwhile that NBS has improved the identification of MPS patients at an early stage. From this point of view, two-tier NBS should be used to identify patients with MPS and to define the exact incidence of MPS.

## 6. Conclusions

The distribution and incidence of MPS fluctuate depending on geographical and ethnic origin. NBS is one of the promising methods for early diagnosis of MPS, and patients could be diagnosed easily at the first stage of life before any clinical manifestations. Newborn screening is changing our knowledge about the real epidemiology of those MPS disorders under reliable screening programs and contributes to early detection and early treatment of the patients.

## Figures and Tables

**Figure 1 diagnostics-11-00273-f001:**
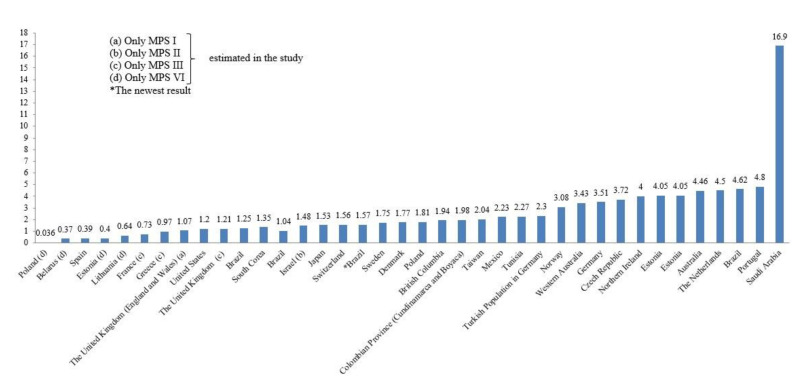
The prevalence of mucopolysaccharidoses per 100,000 live births in 31 countries across the world.

**Figure 2 diagnostics-11-00273-f002:**
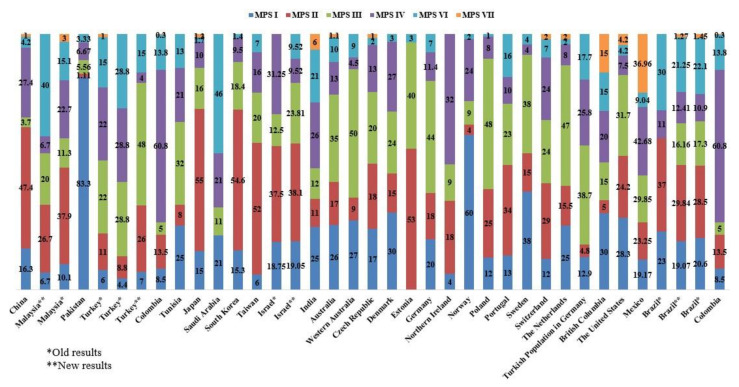
Incidence of MPS (%), Adapted from [1].

**Figure 3 diagnostics-11-00273-f003:**
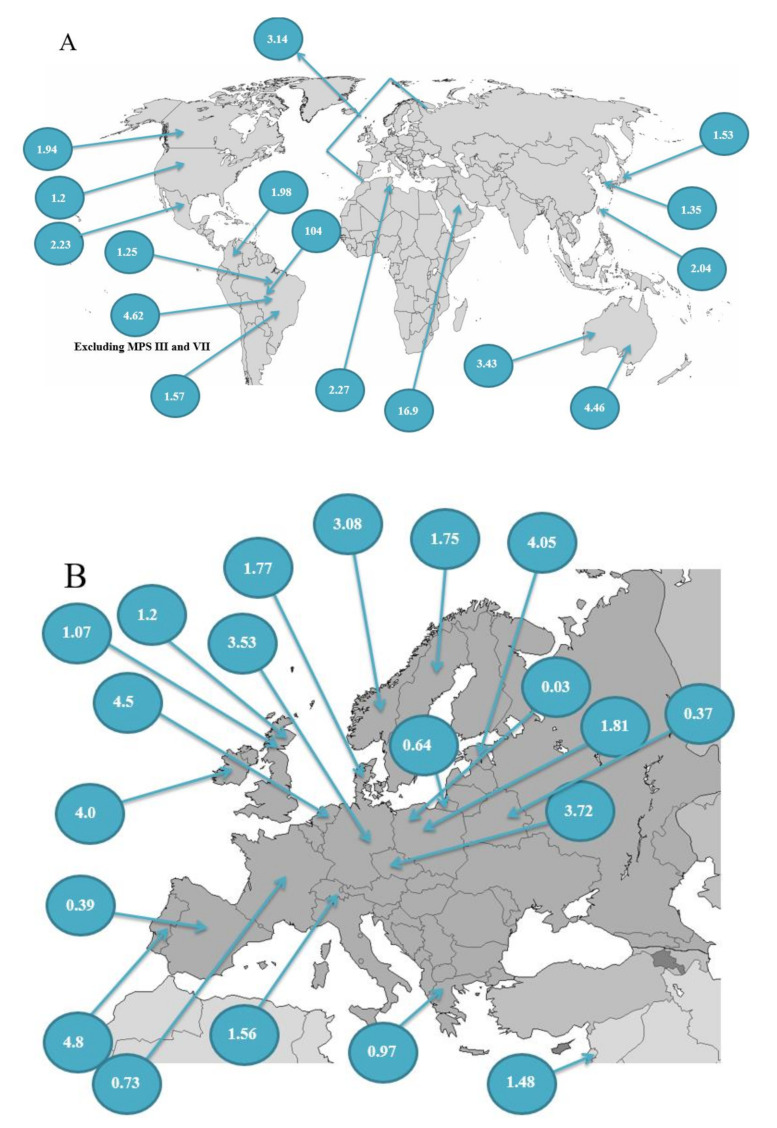
(**A**) Geographic distribution of the combined birth prevalence of MPS; (**B**) Geographic distribution of the combined birth prevalence of MPS in Europe. Adapted from [1].

**Table 1 diagnostics-11-00273-t001:** General characteristics of MPS.

MPS	Associated Gene	Deficient Enzyme	GAG Accumulation	The Severity of Disease	Treatment Options
MPS I-H	IDUA	α-L-iduronidase	HS, DS	Severe	ERT, HSCT
MPS I-S	IDUA	α-L-iduronidase	DS	Mild	ERT, HSCT
MPS I-H/S	IDUA	α-L-iduronidase	DS	Intermediate	ERT, HSCT
MPS IIA	IDS	iduronidase-2-sulfatase	HS, DS	Severe	ERT, HSCT
MPS IIB	IDS	iduronidase-2-sulfatase	HS, DS	Mild	ERT, HSCT
MPS IIIA	SGSH	heparan-N-sulfatase	HS	Variable in the severity	Symptomatic and supportive
MPS IIIB	NAGLU	α-N-acetylglucosaminidase	HS	Variable in the severity	Symptomatic and supportive
MPS IIIC	HGSNAT	α-glucosaminidase acetyltransferase	HS	Variable in the severity	Symptomatic and supportive
MPS IIID	GNS	N-acetylglucosamine-6-sulfatase	HS	Variable in the severity	Symptomatic and supportive
MPS IVA	GALNS	N-acetylglucosamine-6- sulfate sulfatase	C6S, KS	Variable in the severity	ERT, HSCT
MPS IVB	GLB1	ß-galactosidase	KS	Mild	ERT, HSCT
MPS VI	ARSB	N-acetylglucosamine-4-sulfatase	DS, C4S	Variable in the severity	ERT, HSCT
MPS VII	GUSB	ß-glucuronidase	HS, DS, C4S, C6S	Variable in the severity	ERT, HSCT
MPS IX	HYAL1	Hyaluronidase	Hyaluronan	Mild	Symptomatic and supportive

**Table 2 diagnostics-11-00273-t002:** MPS cases and prevalence in different countries by continents.

Country	Study Period	Total MPS Cases	MPS Types						Combined Prevalence	References
			**I**	**II**	**III (A–D)**	**IV (A–B)**	**VI**	**VII**		
**Africa**										
Tunisia	1970–2005	96	0.63	0.29	0.7	0.45	0.3	N/A	2.27	[40]
**Asia**										
Japan	1982–2009	467	0.23	0.84	0.26	0.15	0.03	0.02	1.53	[1]
Saudi Arabia	1983–2008	28	3.62	N/A	1.8	3.62	7.85	N/A	16.9	[36]
South Korea	1994–2013	147	0.21	0.74	0.25	0.13	0.019	N/A	1.35	[37]
Taiwan	1984–2004	130	0.11	1.07	0.39	0.33	0.14	N/A	2.04	[23]
Israel	1967–1975	8	N/A	1.48	N/A	N/A	N/A	N/A	N/A	[20]
**Australia**										
Australia	1980–1996	188	1.14	0.74	1.51	0.59	0.43	0.047	4.46	[20]
Western Australia	1969–1996	18	0.94	0.31	1.71	0.16	0.31	N/A	3.43	[41]
**Europe**										
Belarus	1983–2011	13	N/A	N/A	N/A	N/A	0.37	N/A	N/A	[17]
Czech Republic	1975–2008	119	0.72	0.43	0.91	0.73	0.05	0.02	3.72	[45]
Denmark	1975–2004	33	0.54	0.27	0.43	0.48	0.05	N/A	1.77	[46]
Estonia	1985–2006	15	N/A	2.16	1.62	N/A	0.27	N/A	4.05	[47]
Estonia	1983–2011	2	N/A	N/A	N/A	N/A	0.4	N/A	N/A	[17]
France	1990–2006	128	N/A	N/A	0.73	N/A	N/A	N/A	N/A	[162]
Germany	1980–1995	474	0.69	0.64	1.57	0.38	0.23	N/A	3.51	[48]
Greece	1990–2006	20	N/A	N/A	0.97	N/A	N/A	N/A	N/A	[162]
Lithuania	1983–2011	8	N/A	N/A	N/A	N/A	0.64	N/A	N/A	[17]
Northern Ireland	1958–1985	34	1.66	0.71	0.36	1.3	N/A	N/A	4	[49]
Norway	1979–2004	45	1.85	0.13	0.27	0.76	0.07	N/A	3.08	[46]
Poland	1970–2010	392	0.22	0.46	0.86	0.14	0.0132	N/A	1.8	[50]
Poland	1983–2011	5	N/A	N/A	N/A	N/A	0.036	N/A	N/A	[17]
Portugal	1982–2001	353	1.33	1.09	0.84	0.6	0.42	N/A	4.8	[163]
Sweden	1975–2004	52	0.67	0.27	0.67	0.07	0.07	N/A	1.75	[46]
Switzerland	1975–2008	41	0.19	0.46	0.38	0.38	0.11	0.038	1.56	[1]
The Netherlands	1970–1996	331	1.19	0.67	1.89	0.36	0.15	0.24	4.5	[21]
The United Kingdom	1981–2003	167	1.07	N/A	N/A	N/A	N/A	N/A	N/A	[22]
The United Kingdom	1990–2006	126	N/A	N/A	1.21	N/A	N/A	N/A	N/A	[162]
**North America**										
British Columbia	1952–1986	N/A	0.69	0.9	0.3	0.46	N/A	N/A	N/A	[42]
British Columbia	1969–1996	20	0.58	0.1	0.29	0.39	0.29	0.29	1.94	[43]
The United States	1995–2005	N/A	0.34	0.29	0.38	0.09	0.05	0.05	1.2	[53]
Mexico	2012–2017	198	0.19	0.15	0.17	1.1	0.17	0.23	2.23	[30]
**South America**										
Brazil	1994–2012	600	0.24	0.38	N/A	0.11	0.31	N/A	1.04	[164]
Brazil	1994–2015	823	0.24	0.37	0.21	0.14	0.28	0.02	1.25	[6]
Brazil	1982–2019	1652	0.29	0.48	0.06	0.07	0.35	0.02	1.57	[25]
Colombian province (Cundinamarca and Boyacá)–Colombia	1998–2007	35	0.45	0.45	0.17	0.68	0.23	N/A	1.98	[165]

**Table 3 diagnostics-11-00273-t003:** Incidence of MPS in 29 countries (%).

	Study Period	Total MPS Cases	MPS I	MPS II	MPS III	MPS IV	MPS VI	MPS VII	References
China	2006–2012	190	16.3	47.4	3.7	27.4	4.2	1	[38]
Malaysia	2014–2016	15	6.7	26.7	20	6.7	40	N/A	[29]
Malaysia	N/A	79	10.1	37.9	11.3	22.7	15.1	3	[28]
Pakistan	2013–2015	90	83.3	1.11	5.56	6.67	3.33	N/A	[17]
Turkey	2009–2011	339	6	11	22	22	15	1	[35]
Turkey	N/A	45	4.4	8.8	28.8	28.8	28.8	N/A	[34]
Turkey	N/A	27	7	26	48	4	15	N/A	[33]
Tunisia	1970–2005	96	25	8	32	21	13	N/A	[40]
Japan	1982–2009	467	15	55	16	10	1.7	1.3	[1]
Saudi Arabia	1983–2008	28	21	N/A	11	21	46	N/A	[36]
South Korea	1994–2013	147	15.3	54.6	18.4	9.5	1.4	N/A	[37]
Taiwan	1984–2004	130	6	52	20	16	7	N/A	[23]
Israel	1973	16	18.75	37.5	12.5	31.25	N/A	N/A	[20]
Israel	1974–1979	21	19.05	38.1	23.81	9.52	9.52	N/A	[20]
India	2002–2012	85	25	11	12	26	21	6	[39]
Australia	1980–1996	188	26	17	35	13	10	1.1	[20]
Western Australia	1969–1996	18	27	9	50	4.5	9	N/A	[41]
Czech Republic	1975–2008	119	17	18	20	13	2	1	[45]
Denmark	1975–2004	33	30	15	24	27	3	N/A	[46]
Estonia	1985–2006	15	N/A	53	40	N/A	3	N/A	[47]
Germany	1980–1995	474	20	18	44	11.4	7	N/A	[48]
Northern Ireland	1958–1985	34	4	18	9	32	N/A	N/A	[49]
Norway	1979–2004	45	60	4	9	24	2	N/A	[46]
Poland	1970–2010	392	12	25	48	8	1	N/A	[50]
Portugal	1982–2001	353	13	34	23	10	16	N/A	[19]
Sweden	1975–2004	52	38	15	38	4	4	N/A	[46]
Switzerland	1975–2008	41	12	29	24	24	7	2	[1]
The Netherlands	1970–1996	331	25	15.5	47	8	2	2	[88]
Turkish Population in Germany	1980–1995	62	12.9	4.8	38.7	25.8	17.7	N/A	[48]
British Columbia	1969–1996	20	30	5	15	20	15	15	[43]
The United States	1995–2005	N/A	28.3	24.2	31.7	7.5	4.2	4.2	[53]
Mexico	2012–2017	198	19.17	23.25	29.85	42.68	9.04	36.96	[30]
Brazil	1994–2012	600	23	37	N/A	11	30	N/A	[164]
Brazil	1994–2015	823	20.6	28.5	17.3	10.9	22.1	1.45	[6]
Brazil	1982–2019	1652	19.07	29.84	16.16	12.41	21.25	1.27	[25]
Colombia	1995–2016	319	8.5	13.5	5	60.8	13.8	0.3	[26]

**Table 4 diagnostics-11-00273-t004:** Newborn screening results in seven countries for MPS I–VII.

	Country	Years	Number of DBS Screened	Positive Screened Result	Confirmation Method	Positive	Carrier	Pseudodeficiency	Frequency per 100,000 Birth	References
**MPS I**	Belgium	2015–2016	20,000	54	Liquid chromatography-tandem mass spectrometry	NA	NA	NA	NA	[166]
	Brazil	No date	10,567	2	DNA sequence analysis	0	1	1	0	[167]
	Italy—Umbria	No date	3403	13	Enzyme assay	3	NA	NA	0	[168]
	North East Italy	2015-2017	44,411	13	Tandem mass spectrometry	1	2	5	1/44,411	[169]
	Japan	2012–2015	18,222	300	Liquid chromatography-tandem mass spectrometry	0	0	0	0	[157]
	Mexico	2012–2016	20,018	72	Leukocyte enzyme activity and DNA sequence analysis	2	0	NA	9.99	[161]
	Taiwan	2008–2013	35,285	58	DNA sequence analysis	2	1	NA	5.67	[170]
	Taiwan	2012–2013	60,473	61	DNA sequence analysis	0	NA	NA	0	[171]
	Taiwan	2015–2017	294,196	84	DNA sequence analysis	4	0	0	1.35	[172]
	Taiwan	2015–2017	130,237	120	DNA sequence analysis	5	0	0	3.8	[173]
	Taiwan	2018–2019	73,743	178	UPLC-MS/MS	1	NA	NA	1/73,743	[174]
	USA—Washington	No date	43,000	NA	Mass spectrometry and fluorometric assay	6	NA	NA	13.95	[175]
	USA—Washington	No Date	106,526	9	Tandem mass spectrometry	3	1	NA	8.44 (1/35,700)	[52]
	USA—Kentucky	2016–2017	55,161	76	Tandem mass spectrometry	1	NA	NA	1.81	[176]
	USA—Illinois	2014–2016	219,973	151	Tandem mass spectrometry	1	5	30	1/219,973	[159]
	USA—Missouri	2013–2014	174,636	70	Tandem mass spectrometry	1	3	25	1/174,636	[51]
	USA—Missouri	2013–2017	308	133	Enzyme assay	2	NA	71	0.64	[177]
	USA—Missouri	January–June 2013	43,701	32	Enzyme assay	1	2	7	2.28	[178]
	USA—New York	2015	35,816	13	Liquid chromatography-tandem mass spectrometry and DNA sequence analysis	0	4	8	0	[160]
	USA—New York	No date	43,000	6	Mass spectrometry and fluorometric assay	NA	2	NA	13.6	[175]
**MPS II**	USA—Illinois	2014	93,219	9	Tandem mass spectrometry	1	3	5	NA	[179]
	USA—Washington	No date	105,214	25	Liquid chromatography-tandem mass spectrometry and DNA sequence analysis	1	2	NA	NA	[180]
	Taiwan	2015–2017	294,196	84	Leukocyte enzyme assay and DNA sequence analysis	3	0	0	1.96	[172]
	Taiwan	2018–2019	73,743	56	UPLC-MS/MS	3	NA	NA	1/24,581	[174]
	Taiwan	August to December 2015	28,799	53	DNA sequence analysis	0	NA	NA	NA	[181]
		January 2016–August 2017	101,376	184	DNA sequence analysis	0	NA	38		[181]
**MPS IIIB**	USA—Washington	No date	103,001	0	Liquid chromatography-tandem mass spectrometry and DNA sequence analysis	0	0	0	0	[180]
	Taiwan	2018–2019	73,743	14	UPLC-MS/MS	3	NA	NA	1/24,581	[174]
**MPS IVA**	USA—Washington	No date	106,106	8	Liquid chromatography-tandem mass spectrometry and DNA sequence analysis	0	0	0	0	[180]
	Taiwan	2018–2019	73,743	70	UPLC-MS/MS	6	NA	NA	1/12,291	[174]
**MPS VI**	USA—Washington	No date	103,259	4	Liquid chromatography-tandem mass spectrometry and DNA sequence analysis	0	0	0	0	[180]
	Taiwan	August to December 2015	131,075	176	DNA sequence analysis	0	NA	NA	0	[173]
	Taiwan	2018–2019	73,743	11	UPLC-MS/MS	0	NA	NA	NA	[174]
**MPS VII**	USA—Washington	No date	94,931	1	Liquid chromatography-tandem mass spectrometry and DNA sequence analysis	0	0	0	0	[180]

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
