# Peer review of "Epidemiology of Mucopolysaccharidoses Update"

_diagnostics, 2021, doi:10.3390/diagnostics11020273_

Round 1

Reviewer 1 Report

This is a comprehensive “reference” article with the function combining a catalogue and a tome on epidemiology of the mucopolysaccharidoses (MPS). It showed a quite ambitious goal of covering almost all the aspects of updated epidemiologic issues on MPS and it did the work quite well. However, the way it’s been written needs to be adjusted to make it more acceptable and easy to follow. Also, the authors have to make much clearer statement on the uniqueness of their study and the difference between this manuscript and the article written by Khan et al. in Mol Genet Metab 2017 (not 2018) Jul;121(3):227-240. Since effective treatment methods have been implicated in clinical practice for specific MPS disorders, it is important to make subgroups in MPS I (IH, IH/S and IS) and II (IIA and IIB) for the reference of planning adequate treatment options.

The prevalence of MPS per 100,000 live births in 31 countries across the world in Figure 1 showing mixed data of different single MPS disorders in 9 countries and all the MPS grouped together in one statistic result in the other countries, and the collection and compilation of old and new data with various perspectives from several countries as shown in Figure 2, are quite confusing and not suitable for comparison. It’s not the fault of the study team, however, the missing data from Africa (except Tunisia) makes not only this “reference” article but also the world literature not a complete one. Nonetheless, the collection and analysis of mutations identified in different MPS disorders in different ethnic regions are comprehensive and very helpful.

High risk screening programs were reported but not consistent. The method of urine samples for total uGAG measurement by using DMB was used for at-risk screening in several articles from different regions to identify MPS patients. A report from Taiwan, At-Risk Population Screening Program for Mucopolysaccharidoses by Measuring Urinary Glycosaminoglycans in Taiwan in Diagnostics 2019, 9(4), 140, was not included. Using total uGAG measurement for high risk screening of MPS is not the best way for the purpose. The results were not sufficient to help with the confirmatory diagnosis. Patients with MPS could be missed, especially for MPS IV, VII and even IX.

The statement of “Epidemiology of MPS greatly contributes to prove the accuracy of newborn screening.” in the Conclusion session may not be proper with the still limited data collected up to present time. Rather it could be stated as “Newborn screening is changing our knowledge about the real epidemiology of those MPS disorders under reliable screening programs and contributes to early detection and early treatment of the patients.”

There are some more points for the authors’ reference to make amendments:

In Lines 60-62  However, the therapeutic efficiency of these options has been influenced by various factors, including MPS types, the severity of the disease, age, socioeconomic status, etc. The authors are suggested to make comments on how to define the pathogenicity of new mutations identified in the asymptomatic newborn screening positive neonates, and the delineation of the correlation of these mutations with the severity of phenotypes.

Lines 62-64  The availability of ERT is limited in practice because of the high cost, limited penetration of blood-brain barrier (BBB) and avascular bone region, and weekly or bi-weekly injection.  Cardiac valves should be included in the relatively refractory tissue.

In 2.1. Prevalence of MPS in Brazil

Subtypes of MPS III and MPS IV were identified in 95.50% and 96.09% of the patient populations, respectively. However, differentiation of MPS IH, IH/S and IS in MPS I, and MPS IIA and IIB in MPS II were not presented. Actually, the differentiation and clarification are also important on the clinical basis.

In 2.6. Prevalence of MPS in Taiwan

In a retrospective study in Taiwan during 21 years between 1996 and 2017, 28 patients were identified with MPS III. Within the MPS III group, MPS IIIB had the highest number, which accounted for 86%. The frequency of MPS IIIA and MPS IIIC were 11% and 3%, respectively. No case of MPS IIID was identified during the study period. It’s a pity that an article from Taiwan published in November, 2020, Survival and diagnostic age of 175 Taiwanese patients with mucopolysaccharidoses (1985–2019) Orphanet Journal of Rare Diseases 2020, volume 15: 314, was not included in this paper. It revealed an observation that the life expectancy of Taiwanese patients with MPS has improved in recent decades and patients are being diagnosed earlier. Also age at diagnosis was positively correlated with life expectancy (p < 0.01). The results of pilot newborn screening programs for MPS I, II, VI, IVA, and IIIB which were progressively introduced in Taiwan from 2016 showed that patients born between 2016 and 2019, up to 94% (16/17) were diagnosed with MPS via the newborn screening programs.

In 2.8. Prevalence of MPS in other countries

2.8.1. Japan

The birth prevalence of MPS in Japan was investigated for 27 years from 1982 to 2009. Three hundred thirty-one patients were diagnosed by the screening of uGAGs between 1982 and 1999, and 136 cases were identified by urine analysis from 2003 to 2009. In 2.8.14. Estonia also used similar approach to make the diagnosis on MPS. As screening of uGAGs or application of urine analysis (?) was the 1st tier test for at risk screening which’s not very sensitive or specific, the diagnostic rate might be underestimated.

In 2.8.7. Tunisia

The methodology of making the diagnosis was not mentioned. And, only 76 out of 132 patients were included in the calculation of birth prevalence. That made the estimation of the prevalence not very reliable.

Author Response

Comments and Suggestions for Authors

This is a comprehensive “reference” article with the function combining a catalogue and a tome on epidemiology of the mucopolysaccharidoses (MPS). It showed a quite ambitious goal of covering almost all the aspects of updated epidemiologic issues on MPS and it did the work quite well. However, the way it’s been written needs to be adjusted to make it more acceptable and easy to follow. Also, the authors have to make much clearer statement on the uniqueness of their study and the difference between this manuscript and the article written by Khan et al. in Mol Genet Metab 2017 (not 2018) Jul;121(3):227-240. Since effective treatment methods have been implicated in clinical practice for specific MPS disorders, it is important to make subgroups in MPS I (IH, IH/S and IS) and II (IIA and IIB) for the reference of planning adequate treatment options.

Response: We appreciate the reviewer’s comments and suggestions to make improvement in our article. We have included following in the text the increased number of patients, changing severity of disease, the presence of new mutations every passing day, insufficient and costly treatment and screening options, new outcomes of disease have appeared in the world. That is why we wanted to include the article written by Khan et al (2017) to compare if there is any changes during these three years from 2017 to 2020. We have revised the text according to reviewers instruction to make it easy to follow, and included MPS subtypes as advised.

The prevalence of MPS per 100,000 live births in 31 countries across the world in Figure 1 showing mixed data of different single MPS disorders in 9 countries and all the MPS grouped together in one statistic result in the other countries, and the collection and compilation of old and new data with various perspectives from several countries as shown in Figure 2, are quite confusing and not suitable for comparison. It’s not the fault of the study team, however, the missing data from Africa (except Tunisia) makes not only this “reference” article but also the world literature not a complete one. Nonetheless, the collection and analysis of mutations identified in different MPS disorders in different ethnic regions are comprehensive and very helpful.

Response: We agree with the reviewers concern. In fact, during our literature review, only few articles were found that highlighted African countries, which are listed below; however, these articles have not enough information on prevalence and incidence of MPS, which are very crucial to demonstrate overall data in Africa except Tunisia. Therefore, we did not include data for Africa.

  1. Clarke LA, Atherton AM, Burton BK et al. (2017) Mucopolysaccharidosis type I newborn screening: best practices for diagnosis and management. J Pediatr 182:363–370
  2. Pollard L, Braddoc S, Christensen K, Boylan D, Heese B. Diagnostic follow-up of 47 infants with a positive newborn screen for Hurler syndrome: identification of four recurrent IDUA sequence changes that significantly reduce enzyme activity. In: APHL meeting, Anaheim, CA. Proceedings of the 2014APHl newborn screening and genetic testing symposium, Anaheim, CA, October 27-30, 2014. 2014. http://www.aphl.org/conferences/Documents/Follow-up-2.pdf. Accessed November 26, 2016.
  3. Burlina AB, Polo G, Salviati L, Duro G, Zizzo C, Dardis A, Bembi B, Cazzorla C, Rubert L, Zordan R, Desnick RJ, Burlina AP. Newborn screening for lysosomal storage disorders by tandem mass spectrometry in North East Italy. J Inherit Metab Dis. 2018 Mar;41(2):209-219. doi: 10.1007/s10545-017-0098-3. Epub 2017 Nov 15. PMID: 29143201.

High risk screening programs were reported but not consistent. The method of urine samples for total uGAG measurement by using DMB was used for at-risk screening in several articles from different regions to identify MPS patients. A report from Taiwan, At-Risk Population Screening Program for Mucopolysaccharidoses by Measuring Urinary Glycosaminoglycans in Taiwan in Diagnostics 2019, 9(4), 140, was not included. Using total uGAG measurement for high risk screening of MPS is not the best way for the purpose. The results were not sufficient to help with the confirmatory diagnosis. Patients with MPS could be missed, especially for MPS IV, VII and even IX.

Response: We have included this article in our manuscript. 

The statement of “Epidemiology of MPS greatly contributes to prove the accuracy of newborn screening.” in the Conclusion session may not be proper with the still limited data collected up to present time. Rather it could be stated as “Newborn screening is changing our knowledge about the real epidemiology of those MPS disorders under reliable screening programs and contributes to early detection and early treatment of the patients.”

Response: Thank you for your suggestion. We have included “Newborn screening is changing our knowledge about the real epidemiology of those MPS disorders under reliable screening programs and contributes to early detection and early treatment of the patients” in the text.

There are some more points for the authors’ reference to make amendments:

In Lines 60-62  However, the therapeutic efficiency of these options has been influenced by various factors, including MPS types, the severity of the disease, age, socioeconomic status, etc. The authors are suggested to make comments on how to define the pathogenicity of new mutations identified in the asymptomatic newborn screening positive neonates, and the delineation of the correlation of these mutations with the severity of phenotypes.

Response: Thank you for your suggestions. We have revised it as follows in lines 983-988;

The pathogenicity of new mutations identified in the asymptomatic newborn screening positive neonates can not be defined only by the molecular analysis; however, the recent reports suggest the clinical severity can be predicted by assaying specific GAGs as a second-tier screening. Therefotre, it is critical to measure both enzyme and primary stored GAGs to define the phenotype (pseudodeficiency, attenuated, or severe) along with the molecular analysis. In general, traditional methods can not identify all patients before the patient becomes symptomatic

Refs

  1. Langan, T. J., Jalal, K., Barczykowski, A. L., Carter, R. L., Stapleton, M., Orii, K., Fukao, T., Kobayashi, H., Yamaguchi, S., & Tomatsu, S. (2020). Development of a newborn screening tool for mucopolysaccharidosis type I based on bivariate normal limits: Using glycosaminoglycan and alpha-L-iduronidase determinations on dried blood spots to predict symptoms. JIMD reports52(1), 35–42. https://doi.org/10.1002/jmd2.12093
  2. Stapleton, M., Kubaski, F., Mason, R. W., Shintaku, H., Kobayashi, H., Yamaguchi, S., Taketani, T., Suzuki, Y., Orii, K., Orii, T., Fukao, T., & Tomatsu, S. (2020). Newborn screening for mucopolysaccharidoses: Measurement of glycosaminoglycans by LC-MS/MS. Molecular genetics and metabolism reports22, 100563. https://doi.org/10.1016/j.ymgmr.2019.1005633.
  3. Herbst ZM, Urdaneta L, Klein T, Fuller M, Gelb MH. Evaluation of Multiple Methods for Quantification of Glycosaminoglycan Biomarkers in Newborn Dried Blood Spots from Patients with Severe and Attenuated Mucopolysaccharidosis-I. Int J Neonatal Screen. 2020 Aug 26;6(3):69. doi: 10.3390/ijns6030069. PMID: 33123640; PMCID: PMC7570209.

Lines 62-64  The availability of ERT is limited in practice because of the high cost, limited penetration of blood-brain barrier (BBB) and avascular bone region, and weekly or bi-weekly injection.  Cardiac valves should be included in the relatively refractory tissue.

Response: We have included it in the text.

In 2.1. Prevalence of MPS in Brazil

Subtypes of MPS III and MPS IV were identified in 95.50% and 96.09% of the patient populations, respectively. However, differentiation of MPS IH, IH/S and IS in MPS I, and MPS IIA and IIB in MPS II were not presented. Actually, the differentiation and clarification are also important on the clinical basis.

Response: We agree with reviewer’s concern to report MPS I as MPS IH, IH/S and IS; however, the article by Josahkian et al. (2020) mentioned MPS III and MPS IV subtypes but did not report MPS I and II subtypes.

In 2.6. Prevalence of MPS in Taiwan

In a retrospective study in Taiwan during 21 years between 1996 and 2017, 28 patients were identified with MPS III. Within the MPS III group, MPS IIIB had the highest number, which accounted for 86%. The frequency of MPS IIIA and MPS IIIC were 11% and 3%, respectively. No case of MPS IIID was identified during the study period. It’s a pity that an article from Taiwan published in November, 2020, Survival and diagnostic age of 175 Taiwanese patients with mucopolysaccharidoses (1985–2019) Orphanet Journal of Rare Diseases 2020, volume 15: 314, was not included in this paper. It revealed an observation that the life expectancy of Taiwanese patients with MPS has improved in recent decades and patients are being diagnosed earlier. Also age at diagnosis was positively correlated with life expectancy (p < 0.01). The results of pilot newborn screening programs for MPS I, II, VI, IVA, and IIIB which were progressively introduced in Taiwan from 2016 showed that patients born between 2016 and 2019, up to 94% (16/17) were diagnosed with MPS via the newborn screening programs.

Response: Thanks for reminding out the mistake to include the latest reference. We have included this reference in our revised manuscript.

In 2.8. Prevalence of MPS in other countries

2.8.1. Japan

The birth prevalence of MPS in Japan was investigated for 27 years from 1982 to 2009. Three hundred thirty-one patients were diagnosed by the screening of uGAGs between 1982 and 1999, and 136 cases were identified by urine analysis from 2003 to 2009. In 2.8.14. Estonia also used similar approach to make the diagnosis on MPS. As screening of uGAGs or application of urine analysis (?) was the 1st tier test for at risk screening which’s not very sensitive or specific, the diagnostic rate might be underestimated.

Response: We agree with reviewer that analysis of uGAG is not very sensitive method and may miss patients with low enzyme activity, pseudodeficiency alleles or unknown significant variants, thus cannot be used to evaluate the prevalence of a country alone. It is important to note that these are not the latest report, and these methods could underestimate the diagnostic rate of MPS. Therefore, newborn screening is important for early diagnosis, and enzyme activity on dried blood spot is important to report the accurate rate of MPS.

In 2.8.7. Tunisia

The methodology of making the diagnosis was not mentioned. And, only 76 out of 132 patients were included in the calculation of birth prevalence. That made the estimation of the prevalence not very reliable.

Response: The reason for this calculation was because of the fact only MPS patients whose diagnosis was based on urinary GAGs’s analysis and/or appropriate enzymatic assay within the period 1988 to 2005 were considered in the calculation of the prevalence. Six families with MPS type I and four with MPS type IV had molecular analysis. It has been paraphrased in our revised manuscript.

Reviewer 2 Report

The manuscript presented by the authors is an extensive review indicating the most recent data on the frequency of one of rare genetic diseases, mucopolysaccharidoses (MPS), in various regions of the world, the number of detected mutations, and newborn screening.

The review will certainly be attractive for scientists working on these diseases, as it indicates the current data on epidemiology, but above all for doctors, supporting the diagnosis of these serious diseases.

The work is well written and brings a lot of data. However, I have a few comments and after appropriate revision the manuscript will be ready for publication.

Abstract:

[line 10] The authors point to the lysosomal enzyme deficiency. Then, they describe the diminished quality of life and life-span of patients. There is no sentence here that indicates how defective enzymes can lead to a malfunction of the organism.

[line 17] Before presenting the purpose of the manuscript, it is worth mentioning at least one sentence whether such epidemiological data have already been published. If so, please indicate what makes this work stand out (it indicates newer data, indicates all types, not individual ones, etc.) (such a summary of the work performed to date on the epidemiology of MPS is nicely described in Introduction in line 70). It is also worth mentioning it in the abstract.

Rest of the manuscript:

I have a few questions about Tables and Figures:

(1) It is necessary to correct the numbering of the tables, because Table 1 appears twice [line 79 and 774], which means that the reader may feel confused.

(2) To Table 1 (which in fact should be Table 2, [line 774]) it is worth adding a column with the overall frequency of MPS occurrence in a given country, and not only for individual types.

(3) It is worth to reformate the headings in Table 1 (which in fact should be Table 2, [line 774]). Maybe it would be better to write 'MPS type' in one row and in the following only numbers that indicate types / subtypes?

(4) I don't quite understand the difference between Tables 1 [line 774] and 2 [line 775]. I can see that the way the data is presented is different (frequency per 100,000 births and% of patients). Is it the same set of data or different? Why were these 39 countries selected for Table 2 [line 775]?

(5) There are no links in the text to the tables and figures presented in the manuscript. Table 1 [line 79] is only mentioned in the chapter content before it is shown [in line 56]. The general rule is that a given figure/table should be mentioned in the text. It is enough to use the sentence 'summarized data on this subject are presented in table X'. This makes the reader know when to read the table. Meanwhile, such indications are missing in the case of Table 1 [line 774], Table 2 [line 775] and all Figures presented. Table 3 [line 776] is mentioned in the text only after the table has been presented (in the discussion chapter [line 778]).

Other comments are technical and easy to correct:

[lines 41 i 938] It would be useful to specify the activities of which enzymes are written (for people working on MPS it is obvious that they are lysosomal enzymes but the manuscript can also be read by other people for whom it will not be clear)

[line 101, 117, 140, 171, 343, 780, 805, 808] It is worth checking if there is a double space between some words

[line 273] unnecessary extra comma

[line 364] No explanation of the abbreviation 'NBS'. The explanation for this abbreviation appears in line 758 which is not correct.

It would also be good to quote a couple of the most recent works: Int J Mol Sci. (2020) 21(4): 1258 (DOI: doi: 10.3390/ijms21041258) and Mol Genet Genomic Med. (2020) 8(9):e1356 (DOI: 10.1002/mgg3.1356).

P.S. I have one more minor point. If the contribution of a given person is high enough to be awarded the authorship, there is no need to mention him/her in the Acknowledgements chapter.

Author Response

Comments and Suggestions for Authors

The manuscript presented by the authors is an extensive review indicating the most recent data on the frequency of one of rare genetic diseases, mucopolysaccharidoses (MPS), in various regions of the world, the number of detected mutations, and newborn screening.

The review will certainly be attractive for scientists working on these diseases, as it indicates the current data on epidemiology, but above all for doctors, supporting the diagnosis of these serious diseases.

The work is well written and brings a lot of data. However, I have a few comments and after appropriate revision the manuscript will be ready for publication.

Abstract:

[line 10] The authors point to the lysosomal enzyme deficiency. Then, they describe the diminished quality of life and life-span of patients. There is no sentence here that indicates how defective enzymes can lead to a malfunction of the organism.

Response: We have included “lysosomal enzyme deficiency or malfunction, which leads to the accumulation of glycosaminoglycans in tissues and organs.”

[line 17] Before presenting the purpose of the manuscript, it is worth mentioning at least one sentence whether such epidemiological data have already been published. If so, please indicate what makes this work stand out (it indicates newer data, indicates all types, not individual ones, etc.) (such a summary of the work performed to date on the epidemiology of MPS is nicely described in Introduction in line 70). It is also worth mentioning it in the abstract.

Response: We have revised line 17 as “in the previous study, Khan et al. (2017) have reported the epidemiology of MPS from 22 countries and 16 regions. In this study, we aimed to update the prevalence of MPS across the world. We have collected and investigated 189 publications related to the prevalence of MPS via PubMed as of December 2020. In total, data from 33 countries and 23 regions were compiled and analyzed.”

Rest of the manuscript:

I have a few questions about Tables and Figures:

(1) It is necessary to correct the numbering of the tables, because Table 1 appears twice [line 79 and 774], which means that the reader may feel confused.

Response: We have corrected line 774, which is now 799 as Table 2.

(2) To Table 1 (which in fact should be Table 2, [line 774]) it is worth adding a column with the overall frequency of MPS occurrence in a given country, and not only for individual types.

Response: We have already mentioned individual and combined prevalence in Table 2.

(3) It is worth to reformate the headings in Table 1 (which in fact should be Table 2, [line 774]). Maybe it would be better to write 'MPS type' in one row and in the following only numbers that indicate types / subtypes?

Response: We have added MPS types in table 2, as advised.

(4) I don't quite understand the difference between Tables 1 [line 774] and 2 [line 775]. I can see that the way the data is presented is different (frequency per 100,000 births and% of patients). Is it the same set of data or different? Why were these 39 countries selected for Table 2 [line 775]?

Response: Table 2 represents the prevalence of 29 countries in different continents; however, Table 3 represents the rates of 29 countries, and the data of these two tables are different as rate and prevalence are a different set of measurement.

(5) There are no links in the text to the tables and figures presented in the manuscript. Table 1 [line 79] is only mentioned in the chapter content before it is shown [in line 56]. The general rule is that a given figure/table should be mentioned in the text. It is enough to use the sentence 'summarized data on this subject are presented in table X'. This makes the reader know when to read the table. Meanwhile, such indications are missing in the case of Table 1 [line 774], Table 2 [line 775] and all Figures presented. Table 3 [line 776] is mentioned in the text only after the table has been presented (in the discussion chapter [line 778]).

Response: We have corrected and included it in the text.

Other comments are technical and easy to correct:

[lines 41 i 938] It would be useful to specify the activities of which enzymes are written (for people working on MPS it is obvious that they are lysosomal enzymes but the manuscript can also be read by other people for whom it will not be clear)

Response: We have included “Therefore, the determination of GAG level, as well as the enzyme activity (the deficient enzymes shown in Table 1) and genotyping, is crucial for the diagnosis of MPS” in line 41, and “Until now, a wide range of methods to measure GAGs and enzyme activity (the deficient enzymes shown in Table 1) have been developed in line 938.

[line 101, 117, 140, 171, 343, 780, 805, 808] It is worth checking if there is a double space between some words

Response: We have corrected it.

[line 273] unnecessary extra comma

Response: We have corrected it.

[line 364] No explanation of the abbreviation 'NBS'. The explanation for this abbreviation appears in line 758 which is not correct.

Response: We have corrected and included it.

It would also be good to quote a couple of the most recent works: Int J Mol Sci. (2020) 21(4): 1258 (DOI: doi: 10.3390/ijms21041258) and Mol Genet Genomic Med. (2020) 8(9):e1356 (DOI: 10.1002/mgg3.1356).

Response: We have already cited this article “Mol Genet Genomic Med. (2020) 8(9):e1356 (DOI: 10.1002/mgg3.1356” in our text and added it in the references section. Version that you had included this article under the reference number 90 and the newest version of article included under the reference number 92.  We have cited the article “Int J Mol Sci. (2020) 21(4): 1258 (DOI: doi: 10.3390/ijms21041258)” in our text.

P.S. I have one more minor point. If the contribution of a given person is high enough to be awarded the authorship, there is no need to mention him/her in the Acknowledgements chapter.

Response: We have revised the manuscript. “Acknowledgments: This work was supported by grant from Center for Biomedical Research Excellence (COBRE, grant number P30GM114736)”.